# Local Patterns Generalize Better for Novel Anomalies

**Yalong Jiang**
Beihang University
`AllenYLJiang@outlook.com`

## Abstract

Video anomaly detection (VAD) aims to identify novel actions or events which are unseen during training. Existing mainstream VAD techniques typically focus on the global patterns with redundant details and struggle to generalize to unseen samples. In this paper, we propose a framework that identifies the local patterns which generalize to novel samples and models the dynamics of local patterns. The capability of extracting spatial local patterns is achieved through a two-stage process involving image-text alignment and cross-modality attention. Generalizable representations are built by focusing on semantically relevant components which can be recombined to capture the essence of novel anomalies, reducing unnecessary visual data variances. To enhance local patterns with temporal clues, we propose a State Machine Module (SMM) that utilizes earlier high-resolution textual tokens to guide the generation of precise captions for subsequent low-resolution observations. Furthermore, temporal motion estimation complements spatial local patterns to detect anomalies characterized by novel spatial distributions or distinctive dynamics. Extensive experiments on popular benchmark datasets demonstrate the achievement of state-of-the-art performance. Code is available at `https://github.com/AllenYLJiang/Local-Patterns-Generalize-Better/`.

## 1 Introduction

Video anomaly detection (VAD) is the task of localizing from videos the events that deviate from regular patterns, such as violence, accidents and other unexpected events. Nowadays, numerous platforms such as CCTVs and UAVs play an increasingly important role in surveillance. However, given the vast volume of video data and the low probability of anomalies, it is impractical for humans to manually detect these events. Additionally, visual data variances and domain differences between normal and anomalous events hinder the effectiveness of detection methods. As a result, VAD has become a significant research topic in weakly supervised or unsupervised learning Gong et al. (2019); Shi et al. (2023b); Chalapathy et al. (2017); Lu et al. (2020); Pang et al. (2020); Lv et al. (2021); Georgescu et al. (2021a); Zaheer et al. (2020b); Ristea et al. (2021); Acsintoae et al. (2021).

Existing main-stream works Li et al. (2022c); Luo et al. (2021a); Georgescu et al. (2021a) for VAD are divided into four categories. The first category of methods detects anomalies by leveraging distinctive spatial and temporal features. These methods include prediction-based ones Luo et al. (2021a); Lv et al. (2021); Lu et al. (2020); Park et al. (2020) and reconstruction-based ones Yang et al. (2023b); Lv et al. (2023); Chang et al. (2020); Liu et al. (2021). To enhance representational capacity, some methods combine multi-grained spatio-temporal representations Zhang et al. (2024) for better discrimination, or integrate various features Georgescu et al. (2021a); Cho et al. (2023) to better align with unseen samples Liu et al. (2022b). The second category involves using Multiple Instance Learning (MIL) to iteratively identify useful data segments and fine-tune models for anomaly detection Cho et al. (2023); Wang et al. (2022a); Li et al. (2022a); Zhu et al. (2022); Liu et al. (2023c). For instance, dynamic clustering techniques adapt model representations to real-time observations Wu et al. (2022); Yang et al. (2022). Prompt-enhanced MIL Chen et al. (2024) integrates semantic priors with visual features for improved modeling of anomalies. However, the generalization ability is still insufficient because background noises lead to inconsistent representations over visual data variances, as is shown by Fig. 1. The third category Liu et al. (2023c)

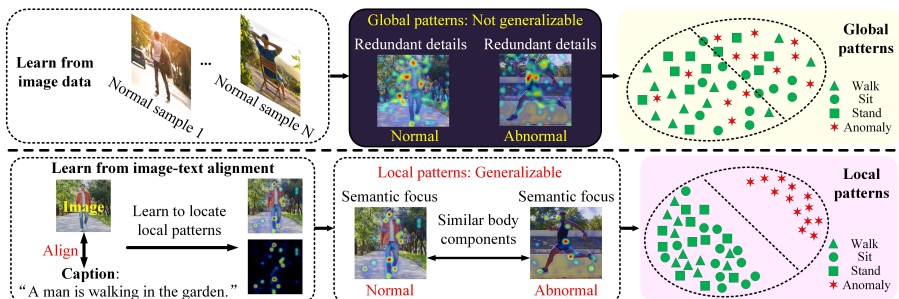

Figure 1: Top: Existing methods rely on global patterns with redundant details, which are inconsistent across visual data variations, limiting their generalization to novel samples. As a result, normal and abnormal samples are poorly distinguished. Bottom: Our method focuses on local patterns that capture semantically meaningful features such as body joints and are consistent across domains and generalize well. The spatial distributions of these local patterns highlight divergences.

focuses on generating realistic anomalies to refine the decision boundary between normal and abnormal samples. Prompt based approaches Wu et al. (2024a) have also been proposed for generating pesudo anomalies. However, the generated anomalies are based on prior assumptions which cannot cover diverse and unexpected anomalous samples in real-world cases. The fourth category Zanella et al. (2024) leverages the visual-language knowledge and reasoning capabilities Yang et al. (2024a) from large models to generate textual descriptions or produce pseudo labels for self-training Yang et al. (2024b), thereby improving the discrimination of abnormal events Micorek et al. (2024).

To generalize model representations to novel anomalies, we propose a two-stage framework for identifying local patterns. In Stage 1, image-text alignment is used to locate text-informative local patterns that are consistent across visual data variances. Stage 2 further refines the local patterns using cross-modality attention, resulting in more compact local patterns. Finally, spatial local patterns are augmented with temporal clues to better determine anomalies.

In sum, the proposed framework is composed of an Image-Text Alignment Module (ITAM) and a Cross-Modality Attention Module (CMAM) for identifying local patterns in two stages. ITAM selects text-informative regions, converting high-dimensional visual data into efficient image tokens. The tokens are converted by Temporal Sentence Generation Module (TSGM) into texts, which CMAM uses to refine the selection of image tokens as local patterns. Temporal clues enhance local patterns in two ways. TSGM generates the sentences for cross-modality attention by considering multi-moment contexts, while temporal motion estimation enriches spatial local patterns with temporal dynamics. The effectiveness is validated on multiple benchmarks, including ShanghaiTech, Ubnormal and so on. The contributions can be highlighted as follows:

- This paper proposes a novel two-stage approach to identify the local patterns that are consistent across visual data variances and generalize to novel abnormal samples. The first stage uses image-text alignment to identify semantically meaningful components, facilitating generalizable representations. Cross-modality attention further refines the components, yielding both the benefits of texts in generalization and the advantages of visual features in representing details.

- Temporal solutions are used to enhance spatial local patterns. Firstly, temporal sentence generation integrates the contexts from different moments to produce coherent descriptions of events. Additionally, temporal motion estimation complements local patterns by modeling dynamics.

- State-of-the-art performance is achieved with the proposed framework on multiple benchmarks.

## 2 RELATED WORKS

### 2.1 UNSUPERVISED VIDEO ANOMALY DETECTION

Due to the unbalanced nature of surveillance videos, most training datasets are without anomaly annotations because it is expensive to label Li et al. (2022b); Liu et al. (2023b); Deng et al. (2023).

Reconstruction-based approaches Astrid et al. (2024); Yang et al. (2023b); Fang et al. (2020); Li et al. (2020a); Gong et al. (2019); Asad et al. (2021); Abati et al. (2019); Sabokrou et al. (2018) produce increased error when encountering irregular features Ramachandra et al. (2020); Madan et al. (2023); Yu et al. (2023) that do not reside in training data. For instance, the method Zaheer et al. (2022a) learns not to reconstruct anomalies. Gong et al. (2019); Gao et al. (2022) augment encoders to improve the sensitivity of reconstruction error to anomalies. Madan et al. (2021); Chang et al. (2020); Singh et al. (2023); Yu et al. (2022b); Shi et al. (2023a) integrate multi-modal features Ding et al. (2021) while Huang et al. (2022) integrates a probabilistic decision model. Zaheer et al. (2022b) assesses the quality of reconstruction to improve stability. Prediction-based methods Luo et al. (2021b); Morais et al. (2019); Luo et al. (2021a); Liu et al. (2018); Nguyen & Meunier (2019); Zeng et al. (2021) evaluate the divergence in normal and abnormal temporal dependencies, leveraging latent spaces Zhang et al. (2020) or hybrid attention Zhang et al. (2022b).

To better distinguish anomalies, Lv et al. (2021); Lu et al. (2020); Liu et al. (2021); Park et al. (2020); Li et al. (2021a) combine prediction with reconstruction. Sato et al. (2023); Wu et al. (2023); Luo et al. (2019) study the distribution over normal samples and propose novel features Arad & Werman (2023). Similarly, Yan et al. (2023) proposes denoising diffusion modules. Flaborea et al. (2023) exploits the enhanced mode coverage of diffusive probabilistic models. To improve representation capacities, Chang et al. (2021); Fan et al. (2024) propose snippet-level attention. Liu et al. (2023a); Yu et al. (2022a); Purwanto et al. (2021) introduce pyramid deformation and CRFs to learn spatio-temporal dependencies Bertasius et al. (2021); Cho et al. (2022). Wang et al. (2021) combines multi-scale features to enhance prediction. Stergiou et al. (2024) combines interpolation with extrapolation for prediction. Wang et al. (2022b) proposes a self-supervised scheme with discriminative DNNs. We propose generalizable local patterns to better represent unseen samples.

## 2.2 Weakly Supervised Anomaly Detection

Multi-instance learning (MIL) takes videos as bags and snippets as instances, transforming video-level labels to instance-level Feng et al. (2021). The methods iteratively locate abnormal segments and fine-tune models using anomalous segments which are dissimilar to normal ones Zhang et al. (2023a). To collect abnormal segments, inter-sample distances are evaluated Lu et al. (2022); Ionescu et al. (2019) based on spatio-temporal similarities Dhiman & Vishwakarma (2020); Lv et al. (2023); Chang et al. (2020); Markovitz et al. (2020). Li et al. (2021b) proposes a probabilistic framework. Sun et al. (2020); Li et al. (2020b) build graphical representations and integrated collective properties in measuring similarities. Sapkota & Yu (2022) performs dynamic non-parametric clustering. To improve robustness, Zhang et al. (2023b) proposes to interpret the vulnerability of MIL. Wu & Liu (2021) introduces causal relations to enhance MIL Tian et al. (2021). Yang et al. (2023a) proposes binary network augmentation strategy. Differently, we propose generalizable representations which facilitate the measurement of similarities between seen and unseen events.

## 2.3 Methods with Data Augmentation

To generate pseudo abnormal samples in fine-tuning, Liu et al. (2023c); Lin et al. (2022); Kim et al. (2022); Liu et al. (2022a); Astrid et al. (2021) propose pseudo abnormal snippet synthesizers which are trained on normal samples Yu et al. (2021). Zaheer et al. (2020a) employs a generator which was not fully trained to create abnormal samples. Chen et al. (2022) generates class balanced training data with a conditional GAN. Lim et al. (2018) focuses on infrequent normal samples during generation, harnessing novel sampling strategies. Besides frame-level analysis Zaheer et al. (2020b), object-level approaches Sun & Gong (2023); Ionescu et al. (2019); Luo et al. (2021a) provide fine-grained analysis. Acsintoae et al. (2022) introduces a new dataset with diverse anomalies. However, the lack in real-world modes in generated data highlights the necessity for generalizable patterns.

## 2.4 Methods Exploring the Representation of Unseen Categories

To adapt model representations and work under changing anomalies, meta learning-based methods Lu et al. (2020); Park et al. (2020), transfer-learning based approaches Doshi & Yilmaz (2020); Perini et al. (2022), continual learning Doshi & Yilmaz (2020) and self-supervised approaches Pang et al. (2020); Degardin & Proença (2021) introduce adaptable feature representations. Attention-based methods Sultani et al. (2018); Guo et al. (2023); Li et al. (2021c); Luo et al. (2017) attend

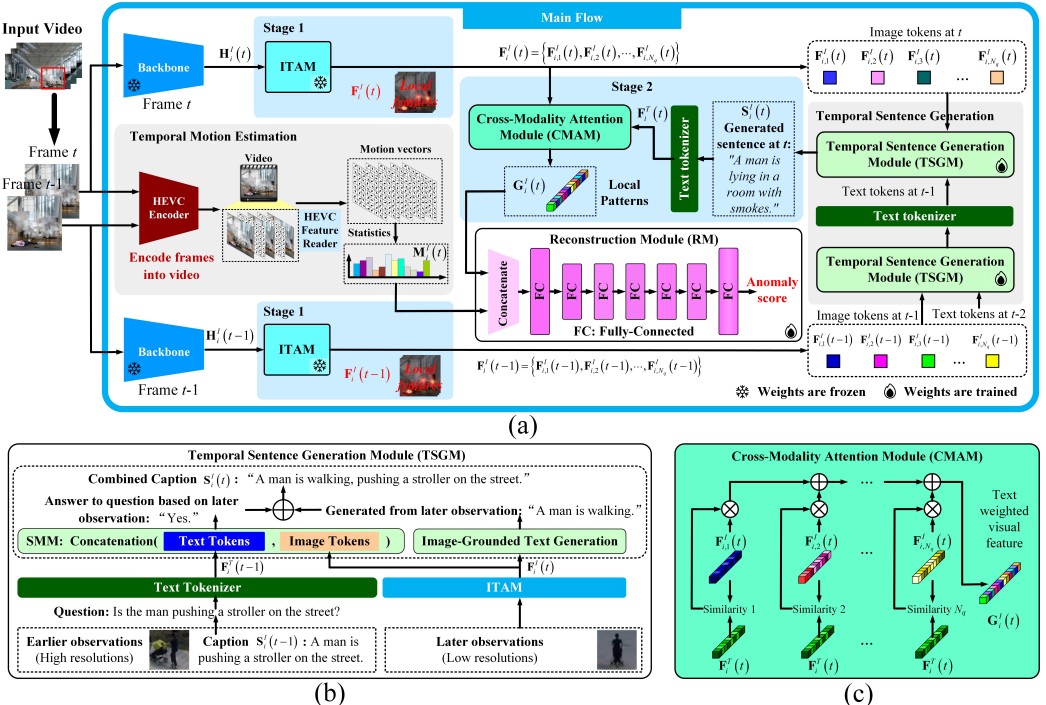

Figure 2: Structure of the method. (a) Main flow: Visual features are extracted using backbone and Image-Text Alignment Module (ITAM) which identifies caption-informative local features as image tokens. Conditioned on image tokens, the Temporal Sentence Generation Module (TSGM) generates sentences which are then combined with image tokens using the Cross-Modality Attention Module (CMAM) to highlight key patterns. HEVC Encoder estimates inter-frame motion through video compression. Spatial local patterns and motion are jointly analyzed to detect anomalies through reconstruction. (b) TSGM uses the State Machine Module (SMM) for sentence generation based on image tokens and earlier sentences. (c) CMAM is implemented based on image-text similarity.

to domain-invariant features in addressing unseen samples. To better align with anomaly detection, Georgescu et al. (2021a) integrates multiple sub-tasks. Zhou et al. (2023a) introduces hierarchical graphs for representing videos and maximizing inter-class margins. Differently, our approach locates the text-informative local patterns which generalize to unseen events.

## 2.5 PROMPTING METHODS

Prompt-based approaches have been widely used in anomaly detection Du et al. (2022); Liu et al. (2023c); Sato et al. (2023). For instance, Zhou et al. (2023b) learns object-agnostic text prompts for generalized abnormality recognition. Yang et al. (2024a) proposes rule-based reasoning to achieve few-normal-shot prompting. Unlike approaches that use direct prompts, we explore local patterns which bridge the gap between images and texts in Visual-Language Models (VLMs).

## 3 METHODOLOGY

To represent unexpected anomalies using generalizable representations, we establish a framework capable of identifying caption-informative local patterns. The framework uses ITAM and CMAM to localize spatial local patterns in two stages, as is shown in Fig. 2(a). To augment local patterns with temporal clues, temporal sentence generation and temporal motion estimation are investigated. Firstly, TSGM models the dependencies between earlier text tokens and later image tokens, enhancing the input sentence for CMAM, as is shown in Fig. 2(b). Then inter-frame motion vectors are obtained from video compression. Finally, spatial and temporal clues are combined in the Reconstruction Module (RM) to detect anomalies. In the following parts we will discuss each module.

## 3.1 Cropping of Image Regions

Due to the wide field of view in some frames containing numerous objects, it is difficult even for GPT-4 Achiam et al. (2023) to focus on all objects together. As a result, local regions are cropped as the first step in our pipeline. We have experimented with both YOLOv7 Wang et al. (2023) and Qwen-7B Bai et al. (2023) for cropping bounding box regions based on prompts. Specifically, "How many people are there?" and "The bounding box of the $i$-th object" are sequentially provided to Qwen-7B which returns corresponding boxes. The comparisons will be included in Appendix D.

## 3.2 Stage 1 for Identifying Spatial Local Patterns

This stage identifies features in cropped image regions that align with texts. The texts describe generic movement attributes (e.g., "A man is walking with swinging arms and legs"). When encountering an unseen action, such as running, the model can recombine known components like arms and legs to generate descriptive language that captures the essence of the action without explicitly naming it. As illustrated in Fig. 1, heatmaps indicate the attention on local components, highlighting similar semantic regions for shared attributes. More visualizations are in Fig. 4.

To identify the local patterns that align with texts, the frozen Image Encoder Li et al. (2023) and the image transformer of Q-Former in BLIP-2 Li et al. (2023) are employed as backbone and ITAM, respectively. The backbone outputs $\mathbf{H}_i^I(t) \in \mathbb{R}^{S_d \times V_d}$, Q-Former has an image transformer and a text transformer for aligning features from both modalities, $\mathbf{F}_i^I(t) \in \mathbb{R}^{N_q \times H_d}$ is the image transformer's output with $N_q$ image tokens $\mathbf{F}_{i,1}^I(t), ..., \mathbf{F}_{i,N_q}^I(t)$ which inform about the captions of image region $i$ and remain consistent over visual data variances, as will be shown by the heatmaps in Fig. 4. Detailed structures are in Appendix C. Algorithm 1 shows the workflow of Stage 1 and Stage 2.

---

**Algorithm 1** Two-Stage Process for Identifying Spatial Local Patterns

---

1: **Input:** Input image, Backbone, ITAM, CMAM, TSGM and Text tokenizer
2: **Output:** Cross-modal embedding $\mathbf{G}_i^I(t)$ representing spatial local patterns
3: **Stage 1: Image Token Extraction**
4:          Use the backbone to extract feature maps $\mathbf{H}_i^I(t)$
5:          Feed $\mathbf{H}_i^I(t)$ into ITAM to obtain image tokens $\{\mathbf{F}_{i,j}^I(t)\}_{j=1}^{N_q}$ which align with texts
6: **Stage 2: Cross-Modality Attention**
7:          Feed image tokens $\{\mathbf{F}_{i,j}^I(t)\}_{j=1}^{N_q}$ into TSGM and obtain text tokens $\mathbf{F}_i^T(t)$
8:          CMAM weightedly sums $\{\mathbf{F}_{i,j}^I(t)\}_{j=1}^{N_q}$ according to their similarity with $\mathbf{F}_i^T(t)$
9: **Return:** Return weighted sum $\mathbf{G}_i^I(t)$, representing the cross-modal features

---

## 3.3 Stage 2 for Identifying Local Patterns

This stage further highlights local patterns by generating a sentence conditioned on image tokens and summing them based on their similarities to the generated sentence. Using the image tokens $\mathbf{F}_{i,1}^I(t), ..., \mathbf{F}_{i,N_q}^I(t)$ from Stage 1, TSGM generates a sentence for image region $i$, as is shown in Fig. 2(a). TSGM utilizes SMM for inter-frame caption augmentation and a frozen Q-Former Li et al. (2023) for image-grounded text generation. SMM determines whether previous events still reside in current frame while Q-Former captions current frame. The outputs from SMM and Q-Former are combined to form the augmented sentence $\mathbf{S}_i^I(t)$. Even with incomplete observations at $t$, $\mathbf{S}_i^I(t)$ can recognize previously occurring events from current frame as long as the events still reside.

The embedding $\mathbf{F}_i^T(t) \in \mathbb{R}^{S_l \times H_d}$ of $\mathbf{S}_i^I(t)$, where $S_l = 32$ denotes the maximum number of tokens in one sentence, is provided to CMAM. CMAM uses the first element in $\mathbf{F}_i^T(t)$ as query and the image tokens as keys and values for attention operations, as is illustrated in Fig. 2(c) and Eq. (1):

$$\mathbf{G}_i^I(t) = (\mathbf{F}_i^T(t)[0]\mathbf{F}_i^I(t)^\top)\mathbf{F}_i^I(t), \mathbf{G}_i^I(t) \in \mathbb{R}^{H_d} \tag{1}$$

$\mathbf{F}_i^T(t)$ is obtained by the text transformer in Q-Former Li et al. (2023) with first element $\mathbf{F}_i^T(t)[0]$ representing the whole sentence. Eq. (1) weightedly sums image tokens according to their cosine

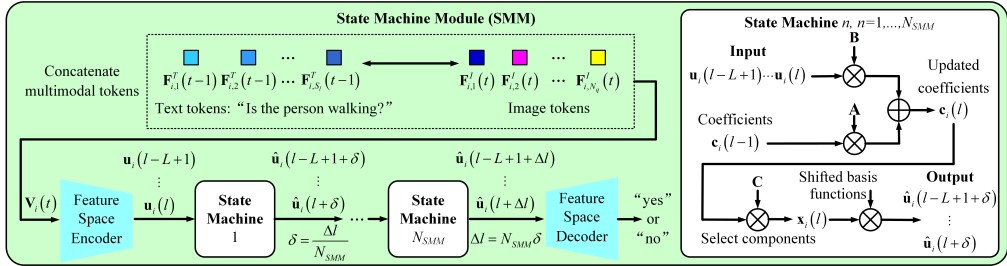

Figure 3: Structure of SMM, which stacks $N_{SMM}$ state machines, each predicting $\delta$ ahead.

similarity to $\mathbf{F}_i^T(t)$. In this way, the advantage of image tokens in characterizing visual details and the benefits of textual features in generalizing over visual data variances are both achieved. Ablation studies will compare the performance of $\mathbf{G}_i^I(t)$ against features from single modalities.

## 3.4 TEMPORAL SENTENCE GENERATION IN STAGE 2

In Stage 2, the generation of captions is influenced by visual data variances such as low resolutions. As is shown in Fig. 2(b), the module Li et al. (2023) for image-grounded text generation only provides a coarse caption "A man is walking" on later low-resolution observations. It is not as precise as earlier caption "A man is pushing a stroller on the street" even if they actually describe the same event. Therefore, SMM in TSGM determines whether earlier high-resolution events are represented by later image tokens. It captures inter-frame dependencies and refines sentence coherence. SMM uses earlier text tokens to generate precise captions for low-resolution observations. The objects in consecutive frames are associated using intersection over union similarity between bounding boxes.

Specifically, SMM augments image tokens $\mathbf{F}_i^I(t) = \{\mathbf{F}_{i,1}^I(t), ..., \mathbf{F}_{i,N_q}^I(t)\}$ with earlier captions $\mathbf{S}_i^I(t-1)$ which are firstly converted from declarative sentences to interrogative sentences. For instance, "The man is pushing a stroller." is changed to "Is the man pushing a stroller?" whose text tokens are $\mathbf{F}_i^T(t-1) = \{\mathbf{F}_{i,1}^T(t-1), ..., \mathbf{F}_{i,S_l}^T(t-1)\}$. Details of this conversion will be shown in Appendix G Hardeniya et al. (2016) . As is shown in Fig. 2(b), SMM combines $\mathbf{F}_i^T(t-1)$ with $\mathbf{F}_i^I(t)$ as input. The state machines in SMM evolve across the dimension of input sequence $\mathbf{V}_i(t) = [\mathbf{F}_{i,1}^T(t-1); ...; \mathbf{F}_{i,S_l}^T(t-1); \mathbf{F}_{i,1}^I(t); ...; \mathbf{F}_{i,N_q}^I(t)]^\top \in \mathbb{R}^{H_d \times (S_l + N_q)}$ where $L = S_l + N_q$ is the sequence length and each token has dimension $H_d$. SMM predicts a binary decision ("yes" or "no") based on the sequence, determining whether the event in $\mathbf{S}_i^I(t-1)$ is still present in $\mathbf{F}_i^I(t)$.

$\mathbf{V}_i(t)$ is deemed as the combination of $H_d$ 1-dimensional signals each with length $L$. The dependencies in sequences are represented using $O$ length-$L$ Legendre polynomials Arfken et al. (2011) $[g_o(1), ..., g_o(L)], o \in [1, O]$, as will be shown in Fig. 6 of Appendix A. The input tensor $\mathbf{V}_i(t)$ is approximated by the weighted sums of the $O$ fixed polynomials. For simplicity, index $t$ is omitted in the following parts which conduct analysis along the column dimension of input tensor at any moment $t$. The Feature Space Encoder produces $\mathbf{U}_i(t) = [\mathbf{u}_i(l - L + 1); ...; \mathbf{u}_i(l)]^\top \in \mathbb{R}^{O \times L}$, where $\mathbf{u}_i(l') = [u_{i,1}(l'), ..., u_{i,O}(l')]$ for $l' \in (l - L, l]$. Here, $l$ varies along the column dimension of $\mathbf{V}_i(t)$ and $\mathbf{U}_i(t)$, $(l - L, l]$ is the window of columns which are encoded by $\mathbf{c}_i(l)$ together.

To better model multi-modal sequence of $H_d$-dimensional signals, $N_{SMM}$ state machines are stacked in SMM, each predicting $\Delta l / N_{SMM}$ ahead, as is shown in Fig. 3. The advantages will be shown in ablation studies. Eq. (2) shows the representation of $\mathbf{U}_i(t)$ with basis functions:

$$u_{i,o}(l') = c_{i,o}(l)g_o(l' - l + L), o \in [1, O], l' \in (l - L, l] \tag{2}$$

In SMM, a state vector $\mathbf{c}_i(l) = [c_{i,1}(l); ...; c_{i,O}(l)]^\top$ with $O$ weights encoding the dependencies between texts and visual tokens in $\mathbf{V}_i(t)$, the dependencies are decomposed onto weighted Legendre basis functions. State vector evolution informs about the prediction ("yes" or "no").

$$\mathbf{c}_i(l + 1) = \mathbf{A}\mathbf{c}_i(l) + \mathbf{B}\sum_{o=1}^{O} u_{i,o}(l + 1) \tag{3}$$

where $\mathbf{A} = \mathbf{A}(O, L) \in \mathbb{R}^{O \times O}$ and $\mathbf{B} = \mathbf{B}(O, L) \in \mathbb{R}^{O \times 1}$ are derived from Legendre polynomials Gu et al. (2020). As $O$ grows, more diversified basis functions can represent more complex dependencies. Assume that $\mathbf{c}_i(l)$ encodes $\mathbf{u}_i(l - L + 1), ..., \mathbf{u}_i(l)$ based on which $\mathbf{u}_i(l + 1)$ is predicted. $\mathbf{u}_i(l+1)$ denotes "yes" or "no". $\mathbf{c}_i(l+1)$ encodes $\mathbf{u}_i(l-L+2), ..., \mathbf{u}_i(l+1)$. Eq. (3) will be derived in Appendix A. In SMM shown by Fig. 3, the transformation $\mathbf{x}_i(l) = \mathbf{C}\mathbf{c}_i(l)$, where $\mathbf{C} \in \mathbb{R}^{O \times O}$ is learnable, highlights important components, Eq. (3) is transformed to

$$\mathbf{x}_i(l) = \mathbf{C}\mathbf{A}^{L-1}\mathbf{B}\sum_{o=1}^{O} u_{i,o}(l - L + 1) + ... + \mathbf{C}\mathbf{B}\sum_{o=1}^{O} u_{i,o}(l) \qquad (4)$$

Finally, the elements of $\mathbf{x}_i(l)$ are multiplied with shifted basis functions $[g_o(1+\delta), ..., g_o(L+\delta)], o \in [1, O], \Delta l = 1, \delta = \Delta l / N_{SMM}$, producing shifted weighted basis functions:

$$\hat{u}_{i,o}(l' + \delta) = x_{i,o}(l)g_o(l' - l + L + \delta), o \in [1, O], l' \in (l - L, l) \qquad (5)$$

The Feature Space Decoder projects $\hat{\mathbf{u}}_i(l + \delta) = [\hat{u}_{i,1}(l + \delta), ..., \hat{u}_{i,O}(l + \delta)]$ onto a prediction ("yes" or "no"), as is shown by Fig. 3. Cross entropy loss is employed. In each batch, $B_s$ images correspond to $B_s$ declarative sentences which are converted into $B_s$ questions. The tokens of each image are concatenated with those of each corresponding question before feeding into SMM.

$$L_{SMM} = -\sum_{i=0}^{B_s-1}\sum_{j=0}^{B_s-1} y_{i,j}log(\frac{Sim(P(i,j), Emb("yes"))}{Sim(P(i,j), Emb("yes")) + Sim(P(i,j), Emb("no"))}) \qquad (6)$$

where ground truth $y_{i,j}$ takes 1 when the Qwen-Chat model Bai et al. (2023) receives question $j$ together with image $i$ and returns "yes", else $y_{i,j}$ takes 0. $Sim(P(i,j), Emb("yes"))$ is the cosine similarity between the embedding $P(i,j)$ of SMM's output and the embedding of "yes".

## 3.5 TEMPORAL MOTION ESTIMATION AND SPATIO-TEMPORAL ANOMALY DETECTION

To enhance the spatial local patterns obtained from Stage 2, this paper proposes to encode frames into H.265 (HEVC) videos using FFmpeg Zeng et al. (2016). As is illustrated in Fig. 2(a), motion vectors from encoded videos are extracted, each motion vector is associated with a $8 \times 8$ macroblock. The orientation of each motion vector is computed as $atan2(y, x)$ and quantized into $D_m = 8$ equi-spaced bins, $x$ and $y$ are the horizontal and vertical components. The average magnitudes of motion vectors in these bins produce a $D_m$-dimensional histogram $\mathbf{M}_i^I(t)$ representing region $i$.

To detect anomalies with anomalous local patterns or irregular dynamics, the Reconstruction Module (RM) with 7 fully-connected layers is trained on normal spatial and temporal data. As is shown in Fig. 2(a), the first layer takes in the concatenation of local patterns $\mathbf{G}_i^I(t)$ and dynamics $\mathbf{M}_i^I(t)$, it maps $H_d + D_m$ input channels to $D_h$ output channels while the last layer maps $D_h$ input channels to $H_d + D_m$ output channels. The 5 hidden layers have $D_h$ input channels and $D_h$ output channels. The reconstructions of spatial and temporal features are conducted together, facilitating the reconstruction of each one to depend on the other. Reconstruction error determines anomaly scores.

## 4 EXPERIMENTS AND RESULTS

This section compares the proposed method with state-of-the-art ones and presents ablation studies.

## 4.1 EXPERIMENTAL SETUP

**Datasets** Experiments are conducted on seven datasets. The training sets of ShanghaiTech, Avenue and UCSD Ped2 contain only normal events and anomalies reside in test data. (1) **ShanghaiTech** dataset Liu et al. (2018) includes 330 training videos and 107 test videos. Among the two versions of ShanghaiTech dataset Liu et al. (2018) and Zhong et al. (2019); Li et al. (2022a); Zanella et al. (2023), the latter includes abnormal behaviors in both training set and test set. As our approach is unsupervised, we use the first version. (2) **CUHK Avenue** dataset Lu et al. (2013) involves 16

Table 1: Performance (AUC, %) on the benchmarks. ST, Ave, UB, Ped2 and NWPU represent ShanghaiTech, CUHK Avenue, Ubnormal, UCSD Ped2 and NWPU Campus, respectively. Macro-AUC and micro-AUC Reiss & Hoshen (2022) are evaluated.

| Algorithm | Year | ST | Ave | UB | Ped2 | NWPU |
|---|---|---|---|---|---|---|
| Georgescu et al. (2021b) | 2021 | 89.3 / 82.7 | 92.3 / 90.4 | - / 61.3 | 99.7 / 98.7 | - |
| Acsintoae et al. (2021) | 2021 | 90.5 / - | 93.2 / - | - | - | - |
| Cai et al. (2021) | 2021 | - / 73.7 | - / 86.6 | - | - / 96.6 | - / 64.5 |
| Reiss & Hoshen (2022) | 2022 | 89.6 / 85.9 | 96.2 / 93.3 | - | **99.9 / 99.1** | - |
| Zhong et al. (2022) | 2022 | - / 74.5 | - / 89.0 | - | - / 98.1 | - |
| Zhang et al. (2022a) | 2022 | - / 80.3 | - / 80.5 | - | - / 92.9 | - |
| Lu et al. (2022) | 2022 | 85.9 / 77.6 | 88.6 / 87.4 | - | - | - / 62.2 |
| Acsintoae et al. (2022) | 2022 | 90.5 / 83.7 | 93.2 / 93.0 | - | - | - |
| Liu et al. (2023c) | 2023 | 91.4 / 85.0 | 93.9 / 93.6 | - | - | - |
| Cao et al. (2023) | 2023 | - / 79.2 | - / 86.8 | - | - | - / 68.2 |
| Hirschorn et al. (2023) | 2023 | - / 85.9 | - | - / 79.2 | - | - |
| Arad & Werman (2023) | 2023 | - / 85.9 | - / 93.5 | - | - / 99.1 | - |
| Sun & Gong (2023) | 2023 | - / 83.4 | - / 93.7 | - | - / 98.1 | - |
| Liu et al. (2023a) | 2023 | - / 78.8 | - / 92.8 | - | - / 99.7 | - |
| Yu et al. (2022a) | 2023 | - / 72.6 | - / 90.7 | - | - / 97.2 | - |
| Zhang et al. (2024) | 2024 | 93.0 / 87.5 | 94.5 / 94.3 | - | - | 72.2 / 70.1 |
| Micorek et al. (2024) | 2024 | 91.5 / 86.7 | 96.1 / 94.3 | 85.5 / 72.8 | 99.9 / 99.7 | - |
| Astrid et al. (2024) | 2024 | - / 71.39 | - / 82.14 | - | - / 94.05 | - |
| Yang et al. (2024a) | 2024 | - / 85.2 | - /89.7 | - / 71.9 | - / 97.9 | - |
| Proposed Method | 2024 | **92.7 / 88.9** | **94.9 / 94.5** | **86.8 / 81.5** | 99.8 / 99.1 | **73.5 / 71.6** |

training videos and 21 test videos. (3) **Ubnormal** dataset Acsintoae et al. (2022) is divided into a training set with 268 videos, a validation set with 64 videos, and a test set with 211 videos. (4) **NWPU Campus** dataset Cao et al. (2023) comprises 43 scenes, 28 classes of anomalies and 16 hours of video footage. (5) **UCSD Ped2** dataset Li et al. (2014) contains 16 normal training videos and 12 test videos. (6) **UCF Crime** dataset Sultani et al. (2018) includes 1610 training videos in which 800 contain only normal behaviors. The test set includes 290 videos in which 140 include anomalies. (7) **XD Violence** Wu et al. (2020) includes 4754 videos where 2349 are non-violent and 2405 are violent. There are 3954 training videos and 800 test videos where 500 are violent.

**Evaluation Metrics** Following previous literature Markovitz et al. (2020), Area under Curve (AUC, %) is adopted for evaluation. Differently, the accuracy on XD-Violence dataset is measured using precision-recall curve and the corresponding Average Precision (AP, %) Panariello et al. (2022).

**Implementation Details** To capture more contexts, bounding boxes are expanded by 50% on both sides horizontally and vertically. The benefits of box expansion will be shown in Table 4 of Appendix D. For image region $i$ at $t$, the output of backbone and ITAM are $\mathbf{H}_i^I(t) \in \mathbb{R}^{S_d \times V_d}$ and $\mathbf{F}_i^I(t) \in \mathbb{R}^{N_q \times H_d}$ which satisfy $S_d = 257$, $V_d = 1408$, $N_q = 32$, $H_d = 768$. Each of the $N_q$ image tokens has embedding size $H_d$. Following BLIP-2 Li et al. (2023), the backbone has "ViT-L/14" structure in Radford et al. (2021). The text tokenizer in Fig. 2(a) will be detailed in Appendix C. For sentences with fewer than $S_l$ tokens, $\mathbf{F}_i^T(t)$ is padded with zeros. RM has $D_h = 512$ in intermediate layers.

SMM, with $N_{SMM} = 3$ state machines, is trained on the COCO-Caption dataset Lin et al. (2014). Table 3 shows the influences of $N_{SMM}$. The Feature Space Encoder ($H_d$ input channels, $O = 64$ output channels) and Feature Space Decoder ($O$ input channels, $H_d$ output channels) are learnable fully-connected layers, the weights in $\mathbf{C} \in \mathbb{R}^{O \times O}$ are also learnable. All weights are initialized with distribution $N(0, 0.02)$. Training spans 20 epoches with initial learning rate $5 \times 10^{-5}$ and decay 0.99. RM takes concatenated $\mathbf{G}_i^I(t)$ and $\mathbf{M}_i^I(t)$ as input, with ReLU activations. It is trained using Adam optimizer with learning rate $10^{-3}$ for 10 epoches, using MSE loss. Implementations are based on Pytorch Pytorch (2018) and a NVIDIA A100 GPU. RM is trained on benchmark videos without anomalies. The influences of RM's number of layers will be shown in Appendix F. The evaluations on operational efficiency will be detailed in Appendix H.

Table 2: Performance on UCF-Crime (micro-AUC, %) and XD-Violence (AP, %). UCF and XD represent UCF-Crime and XD-Violence, respectively.

| Algorithm | UCF | XD | Algorithm | UCF | XD |
|---|---|---|---|---|---|
| Joo et al. (2023) | 87.58 | 82.19 | Wu et al. (2024a) | 86.40 | 76.03 |
| Chen et al. (2023) | 86.98 | 80.11 | Chen et al. (2024) | 86.83 | 88.21 |
| Pu et al. (2023) | 86.76 | 85.59 | Yang et al. (2024b) | 87.79 | 83.68 |
| Tan et al. (2024) | 86.71 | 82.10 | Wu et al. (2024b) | 88.02 | 84.51 |
| Zanella et al. (2024) | 80.28 | 85.36 | Proposed Method | **88.83** | **86.96** |

Table 3: Ablations of components using Micro-AUC. TME and TSG are short for Temporal Motion Estimation and Temporal Sentence Generation, respectively. $N_q$ is the number of image tokens.

| Setting | Stage 1 | Stage 2 | $N_q$ | TME | TSG | $N_{SMM}$ | ST | Ave | UB | Ped2 |
|---|---|---|---|---|---|---|---|---|---|---|
| 1 | × | × | 32 | × | × | - | 71.9 | 85.1 | 72.8 | 79.3 |
| 2 | ✓ | × | 32 | × | × | - | 80.3 | 86.5 | 73.6 | 90.7 |
| 3 | ✓ | ✓ | 32 | × | ✓ | 3 | 86.4 | 88.7 | 79.3 | 96.8 |
| 4 | ✓ | w/o image tokens | 32 | × | ✓ | 3 | 79.2 | 87.8 | 72.1 | 95.4 |
| 5 | ✓ | w/o text tokens | 32 | × | ✓ | 3 | 80.7 | 89.6 | 74.5 | 95.9 |
| 6 | ✓ | ✓ | 32 | ✓ | ✓ | 3 | 88.9 | 94.5 | 81.5 | 99.1 |
| 7 | ✓ | ✓ | 32 | ✓ | w/o SMM | 3 | 87.6 | 93.0 | 80.1 | 98.5 |
| 8 | × | × | 32 | ✓ | × | - | 84.1 | 86.2 | 77.6 | 94.5 |
| 9 | ✓ | ✓ | 32 | ✓ | ✓ | 1 | 88.6 | 94.4 | 79.8 | 99.1 |
| 10 | ✓ | ✓ | 32 | ✓ | ✓ | 5 | 88.9 | 94.5 | 81.5 | 99.1 |

## 4.2 COMPARISONS WITH BASELINES

To demonstrate the superiority, the proposed approach is compared with existing ones, including LLM-based baselines Yang et al. (2024a), for detecting anomalies. Significant improvements are observed in Table 1. Such improvements are attributed to the identification of spatial local patterns and dynamics. Results on non-human objects are shown in Table 2 with UCF-Crime and XD-Violence datasets, suggesting that local patterns can generalize to different object types.

## 4.3 ABLATION STUDIES

**Ablation on Stage 1 and Stage 2** In Setting 1 of Table 3, the reconstruction error of backbone features $\mathbf{H}_i^I(t)$ is used to detect anomalies. Setting 2 and Setting 3 show the utilization of Stage 1 and both stages for reconstruction, respectively. The comparison shows that Stages 1 and 2 both play crucial roles in selecting text-informative local patterns, as will be illustrated in Fig. 4.

**Ablation on ITAM's Structure** To demonstrate that the primary contributor to generalization is image-text alignment instead of pre-existing models, we conduct an ablation study by varying the structure and training data of ITAM. Detailed results and analysis are provided in Appendix E.

**Ablation on Cross-Modality Attention** Setting 4 in Table 3 replaces cross-modality feature $\mathbf{G}_i^I(t)$ in Setting 3 with textual feature $\mathbf{F}_i^T(t)$ of sentence from TSGM, using reconstruction error on $\mathbf{F}_i^T(t)$ to determine anomalies. Setting 5 discards text tokens, only using the reconstruction error on $\mathbf{F}_i^I(t)$. Fig. 5 also shows that combining visual and textual features outperforms using a single modality.

**Ablation on Temporal Motion Estimation** The improvement of Setting 6 over Setting 3 demonstrates that temporal dynamics complements local patterns in detecting anomalies. Setting 8 shows the performance of only using reconstruction error on dynamics $\mathbf{M}_i^I(t)$ for anomaly detection.

**Ablation on the SMM in Temporal Sentence Generation in Stage 2** The comparison between Setting 6 and Setting 7 shows that if TSGM only uses Q-Former Li et al. (2023) for image-grounded text generation without SMM to incorporate previous captions, performance drops. As a result, the mixture of image tokens and text tokens from different moments contributes to more informative sentences. More ablations on SMM will be shown in Appendix B.

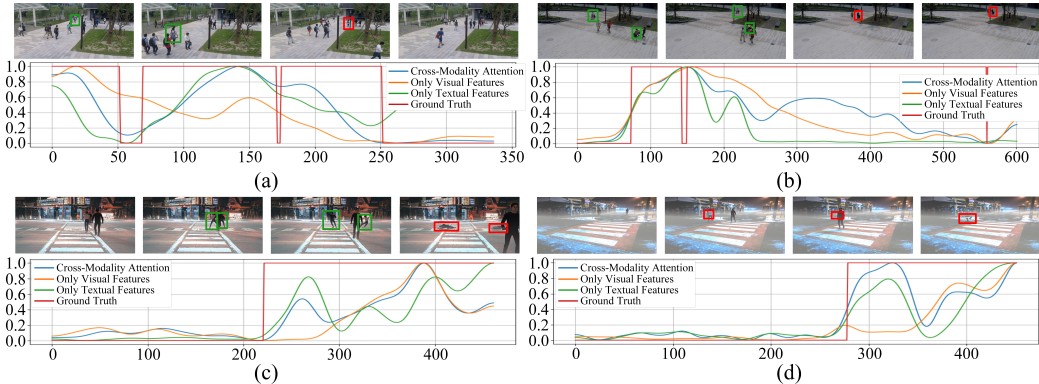

Figure 4: Heatmaps of local patterns in two stages. (a) Normal events. (b) Abnormal events. Both (a) and (b) follow the same row arrangement: the first row contains input images, the second row shows the features from ITAM, and the third row shows the local patterns selected by CMAM.

Figure 5: Anomaly scores obtained using image features, text features and combined ones. Cross-modality attention detects anomalies even under occlusion and low resolution. Green and red boxes show the anomalies detected with single modality and cross-modality attention, respectively.

**Ablation on SMM's Structure** SMM stacks $N_{smm}$ state machines, each predicting a future period of $\Delta l/N_{smm}$. The stacking mechanism achieves the full prediction $\Delta l$. Setting 6, 9 and 10 in Table 3 show that $N_{smm} = 3$ outperforms $N_{smm} = 1$. The task for each state machine becomes simpler because each one focuses on short-term dependencies. Predicting a long period $\Delta l$ requires capturing both short- and long-term dependencies. A single state machine struggles to handle these varying dependencies effectively, especially in our case with non-linear multi-modal dependencies.

Moreover, the ablation on the number of image tokens $N_q$ will be involved in Appendix E.

### 4.4 SUBJECTIVE RESULTS ON LOCAL PATTERNS

Fig. 4 subjectively shows local patterns. The second rows of Fig. 4(a) and (b) highlight the patterns for $\mathbf{F}_i^I(t)$ while the third rows display those for $\mathbf{G}_i^I(t)$. The heatmaps, generated using Grad-CAM Selvaraju et al. (2017), show that local patterns span similar semantic regions across normal and abnormal events. Cross-modality attention refines these patterns to focus on semantically relevant components, enhancing generalization. More visualizations will be presented in Appendix I.

## 5 DISCUSSION AND CONCLUSION

**Limitations:** The limitation of our work lies in the reliance on object detectors, because the direct processing of an image with many objects using VLM can result in context being ignored. Please refer to Appendix J for more potential directions of improvement.

**Conclusions:** In this paper, we establish a framework for video anomaly detection by locating local patterns through image-text alignment and cross-modality attention. At the core of the framework is identifying the text-informative local patterns that generalize to novel anomalies, ensuring consistent representations across novel visual data. Additionally, temporal sentence generation and motion estimation augment cross-modality attention and complement spatial local patterns, respectively. Extensive experiments show that the framework surpasses existing state-of-the-art methods.

ACKNOWLEDGEMENT

This work is supported by the National Natural Science Foundation of China under Grant (62301020) and Beijing Natural Science Foundation under Grant (4234085).

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

## A  DETAILS ABOUT SMM

We model the sequence $\mathbf{V}_i(t) = [\mathbf{F}_{i,1}^T(t-1);...;\mathbf{F}_{i,S_l}^T(t-1);\mathbf{F}_{i,1}^I(t);...;\mathbf{F}_{i,N_q}^I(t)]^\top \in \mathbb{R}^{H_d \times (S_l + N_q)}, L = S_l + N_q$ of object $i$ consisting of both $S_l$ columns denoting the sentence from $t-1$ and $N_q$ columns denoting image tokens at $t$. Matrix $\mathbf{V}_i(t)$ is projected to $\mathbf{U}_i(t) \in \mathbb{R}^{O \times L}$ with the Feature Space Encoder which is a fully-connected layer in Fig. 3. The $l'$-th column in $\mathbf{U}_i(t)$ is $[u_{i,1}(l'),...,u_{i,O}(l')]^\top, l' \in (l-L,l]$. $l$ varies along the column dimension of $\mathbf{V}_i(t)$ and $\mathbf{U}_i(t)$, $(l-L,l]$ is the window of columns which are encoded by $\mathbf{c}_i(l)$ together. For simplicity, we ignore index $t$ in the following parts which conduct analysis along the column dimension of input tensor at any moment $t$.

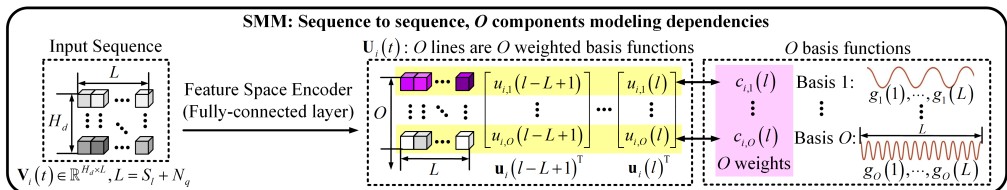

Figure 6: Basis functions in state machines for modeling sequences.

Fig. 6 shows the basis functions in state machines for modeling sequences. The $O$ rows in $\mathbf{U}_i(t)$ are the weighted version of the $O$ length-$L$ Legendre polynomials Arfken et al. (2011) $[g_o(1),...,g_o(L)], o \in [1, O]$. Specifically, the $o$-th row can be represented as:

$$[u_{i,o}(l-L+1),...,u_{i,o}(l)] = c_{i,o}(l)[g_o(1),...,g_o(L)], o \in [1, O] \tag{7}$$

In SMM, a state vector $\mathbf{c}_i(l) \in \mathbb{R}^O$ with $O$ weights $c_{i,1}(t),...,c_{i,O}(t)$ are the weights of polynomials and encode the dependencies among the columns in $\mathbf{V}_i(t)$.

As can be seen from Fig. 6, $\mathbf{c}_i(l_1)$ encodes the dependencies among $l_1 - L + 1,...,l_1$ columns, $\mathbf{c}_i(l_2)$ encodes the dependencies among $l_2 - L + 1,...,l_2$ columns. According to Gu et al. (2020), the dynamics of a 1-dimensional sequence $f_i(l)$ across a period can be represented by $\mathbf{c}_i(l) \in \mathbb{R}^O$, satisfying $[f_i(l-L+1),...,f_i(l)] = \sum_{o=1}^O c_{i,o}(l)[g_o(1),...,g_o(L)], f_i(l') \in \mathbb{R}, l' \in [l-L+1, l]$. The transitions from $\mathbf{c}_i(l_1)$ to $\mathbf{c}_i(l_2)$ facilitates the prediction in the sequence:

$$\frac{d}{dl}\mathbf{c}_i(l) = \mathbf{A}_{HiPPO}\mathbf{c}_i(l) + \mathbf{B}_{HiPPO}f_i(l) \tag{8}$$

By combining Eq. (7) with Eq. (8), we can obtain:

$$\frac{d}{dt}\mathbf{c}_i(l) = \mathbf{A}_{HiPPO}\mathbf{c}_i(l) + \mathbf{B}_{HiPPO}\sum_{o=1}^O u_{i,o}(l) \tag{9}$$

The matrices $\mathbf{A}_{HiPPO}$ and $\mathbf{B}_{HiPPO}$ are defined in Gu et al. (2020) with $o, h \in [1, O]$:

$$\mathbf{A}_{HiPPO}(o, h) = \begin{cases} -\frac{(2o+1)^{0.5}(2h+1)^{0.5}}{L} & \text{if } o > h, \\ 0 & \text{if } o < h, \\ -\frac{o+1}{L} & \text{if } o = h. \end{cases} \tag{10}$$

$$\mathbf{B}_{HiPPO}(o) = -\frac{(2o+1)^{0.5}}{L} \tag{11}$$

To discretize Eq. (9), we obtain

$$\lim_{\Delta \to 0} \frac{\mathbf{c}_i(l+\Delta) - \mathbf{c}_i(l)}{\Delta} = \lim_{\Delta \to 0} \left( \frac{\mathbf{A}_{HiPPO}\mathbf{c}_i(l) + \mathbf{B}_{HiPPO} \sum_{o=1}^{O} u_{i,o}(l)}{2} + \right.$$
$$\left. \frac{\mathbf{A}_{HiPPO}\mathbf{c}_i(l+\Delta) + \mathbf{B}_{HiPPO} \sum_{o=1}^{O} u_{i,o}(l+\Delta)}{2} \right), \Delta = 1 \tag{12}$$

which can be transformed to

$$\mathbf{c}_i(l) = \frac{\mathbf{I} + \frac{\Delta}{2}\mathbf{A}_{HiPPO}}{\mathbf{I} - \frac{\Delta}{2}\mathbf{A}_{HiPPO}} \mathbf{c}_i(l-1) + \frac{\Delta \mathbf{B}_{HiPPO}}{\mathbf{I} - \frac{\Delta}{2}\mathbf{A}_{HiPPO}} \sum_{o=1}^{O} u_{i,o}(l) \tag{13}$$

which simplifies to

$$\mathbf{c}_i(l) = \mathbf{A}\mathbf{c}_i(l-1) + \mathbf{B} \sum_{o=1}^{O} u_{i,o}(l) \tag{14}$$

As a result, $\mathbf{A} = \mathbf{A}_{Legendre}(O, L) \in \mathbb{R}^{O \times O}$ and $\mathbf{B} = \mathbf{B}_{Legendre}(O, L) \in \mathbb{R}^{O \times 1}$ are determined by Legendre bases.

## B  SUBJECTIVE RESULTS OF TEMPORAL SENTENCE GENERATION WITH SMM

In Fig. 7, the green bounding boxes indicate the anomalies that can be detected by directly applying the Q-Former, as described in Li et al. (2023), for image-grounded text generation in TSGM. The red bounding boxes show the cases where only with the combination of SMM and Q-Former in TSGM can the anomalies be detected. The curves show anomaly scores. Under poor observational conditions like occlusions and low resolutions, SMM complements the Q-Former in TSGM to effectively detect abnormal events.

## C  STRUCTURES OF MODULES FOR IMAGE-TEXT ALIGNMENT AND IMAGE-GROUNDED TEXT GENERATION

ITAM is the image transformer of Q-Former Li et al. (2023), as is shown in Fig. 8. It outputs $\mathbf{F}_i^I(t) \in \mathbb{R}^{N_q \times H_d}$ which is aligned with the output from text transformer Li et al. (2023) during training to learn extracting text-aligned features. The text tokenizer in Fig. 2 is part of the text transformer of Q-Former Li et al. (2023). ITAM and text tokenizer are frozen in our work.

**Image Transformer** To select from $\mathbf{H}_i^I(t)$ the caption-informative local patterns, this module is built with self-attention layers, cross-attention layers and feed-forward layers, as is shown by Fig. 8. Firstly, $N_q$ learnable query embeddings attend to each other in self-attention layers before interacting with $\mathbf{H}_i^I(t)$ through cross-attention layers. Each query embedding has dimension $H_d$. The Image-attention Module involves 6 sequential transformer layers each of which includes one self-attention layer, one cross-attention layer and one feed-forward layer. $\mathbf{H}_i^I(t)$ acts as a static input to the cross-attention layers across all transformer layers. The transformer layers sequentially refine the understanding and integration of $\mathbf{H}_i^I(t)$ with learned queries. Each self-attention layer is implemented according to Vaswani et al. (2017) with 12 heads, producing output $\mathbf{Q}_i^I(t) \in \mathbb{R}^{N_q \times H_d}$. Each cross-attention layer has 12 heads with $\mathbf{H}_i^I(t)$ functioning as key and value, it performs feature fusion by combining $\mathbf{H}_i^I(t)$ with $\mathbf{Q}_i^I(t)$ to $\mathbf{Z}_i^I(t) \in \mathbb{R}^{N_q \times H_d}$. $\mathbf{Z}_i^I(t)$ is projected by fully-connected feed-forward layers to $\mathbf{F}_i^I(t) \in \mathbb{R}^{N_q \times H_d}$.

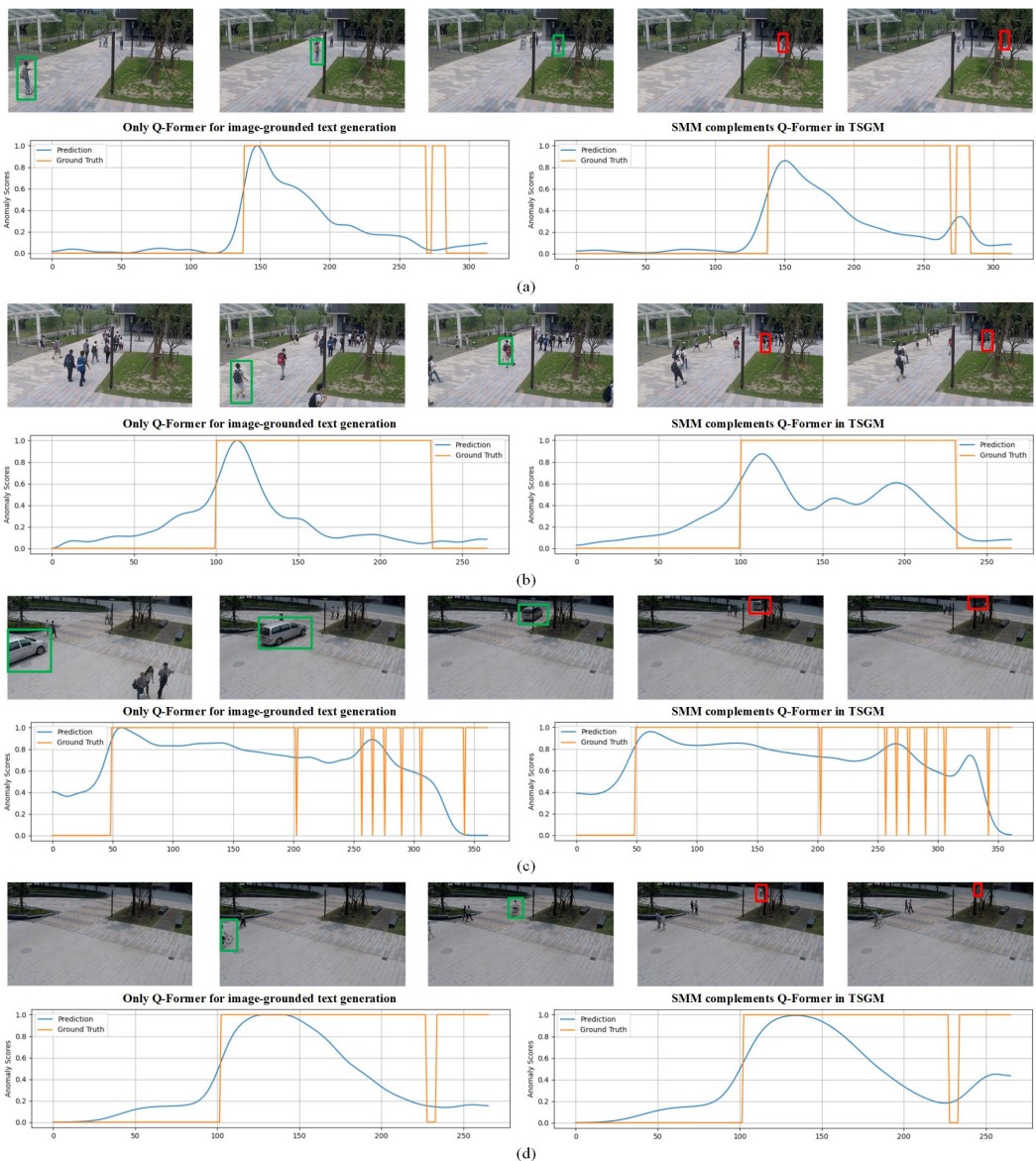

Figure 7: Demonstration of the effectiveness of SMM in TSGM. (a) The man is riding a unicycle but viewed under low resolutions. (b) The man is running but viewed under low resolutions. (c) The vehicle is viewed under occlusions. (d) The man is riding a bicycle but viewed under low resolutions.

**Text Transformer** To encode captions, the module is built with self-attention layers and feed-forward layers, as is shown by Fig. 8. The self-attention layers and feed-forward layers are shared by Image-attention Module and Text-attention Module. In self-attention modules, the text tokens $\mathbf{E}_j^I(t) = [\mathbf{E}_{j,1}^I(t), \mathbf{E}_{j,2}^I(t), ..., \mathbf{E}_{j,S_l}^I(t)] \in \mathbb{R}^{S_l \times H_d}$ in a sentence with maximum length $S_l$ attend to each other. $S_l = N_q$ and $\mathbf{E}_{j,1}^I(t) \in \mathbb{R}^{H_d}, \mathbf{E}_{j,2}^I(t) \in \mathbb{R}^{H_d}, ..., \mathbf{E}_{j,S_l}^I(t) \in \mathbb{R}^{H_d}$.

To shorten the embeddings of an entire sequence, we follow Devlin et al. (2018) by prepending special token [CLS] to the start of input sequence for aggregating information based on the fact that all tokens attend to each other. Due to the fact that the first token informs about the whole sequence, we only keep the first element $\mathbf{F}_i^T(t)[0] \in \mathbb{R}^{H_d}$ of text transformer's output $\mathbf{F}_i^T(t) \in \mathbb{R}^{S_l \times H_d}$.

**Image-Grounded Text Generation Module** Conditioned on visual features $\mathbf{F}_i^I(t)$, the module iteratively generates new text tokens until the full sentence with maximum length $S_l$ is produced.

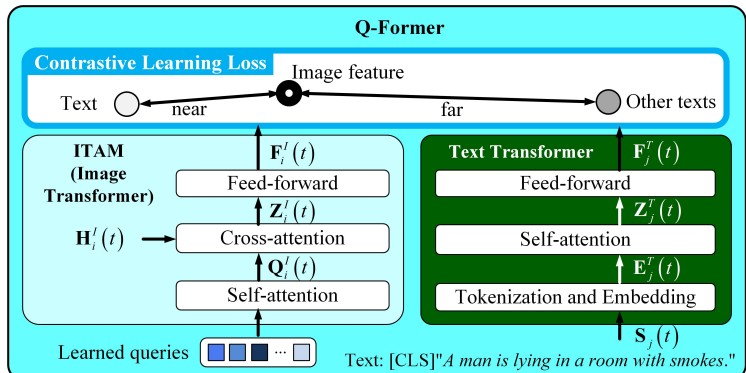

Figure 8: ITAM is the image transformer in Q-Former Li et al. (2023), it identifies the local features through aligning the visual features from image transformer with textual features from text transformer.

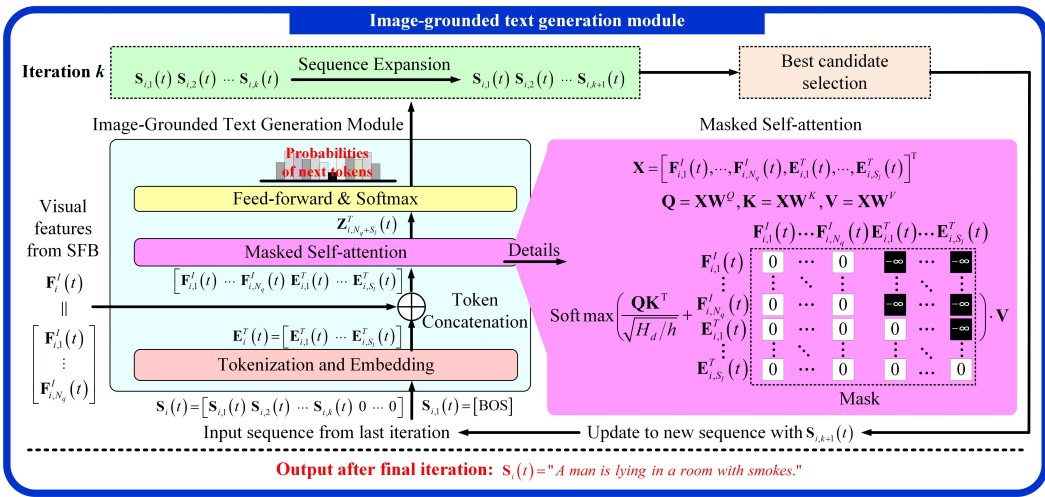

Figure 9: Structure of the module for image-grounded text generation, the module is part of the Q-Former Li et al. (2023).

Following Radford et al. (2018) Devlin et al. (2018) where token "[BOS]" signals the start of text generation, we initialize the sentence to be "[BOS]" following by $S_l - 1$ zero placeholders. In each iteration, a new text token is generated and replaces one zero placeholder, as is shown in Fig. 9.

In the $k - th$ iteration, the input sequence has previously generated tokens $\mathbf{S}_{i,1}(t), \mathbf{S}_{i,2}(t), ..., \mathbf{S}_{i,k}(t)$ followed by $S_l - k$ zero placeholders, producing embeddings $\mathbf{E}_{i,1}^T(t), \mathbf{E}_{i,2}^T(t), ..., \mathbf{E}_{i,S_l}^T(t)$. The visual embeddings are concatenated with text tokens, producing $\mathbf{X} = [\mathbf{F}_{i,1}^I(t), \mathbf{F}_{i,2}^I(t), ..., \mathbf{F}_{i,N_q}^I(t), \mathbf{E}_{i,1}^T(t), \mathbf{E}_{i,2}^T(t), ..., \mathbf{E}_{i,S_l}^T(t)]^T$ as the input to self-attention layer. As is shown in Fig. 9, the mask in self-attention layer enables visual tokens to attend to each other, and facilitates each of the $S_l$ text tokens attend to all visual tokens and earlier text tokens. Specifically, provided query, key and values $\mathbf{Q} = \mathbf{X}\mathbf{W^Q}$, $\mathbf{K} = \mathbf{X}\mathbf{W^K}$ and $\mathbf{V} = \mathbf{X}\mathbf{W^V}$ with $\mathbf{W^Q}$, $\mathbf{W^K}$ and $\mathbf{W^V}$ being learnable weights, self-attention is implemented by

$$\mathbf{Z}_i^T(t) = Softmax(\frac{\mathbf{Q}\mathbf{K^T}}{\sqrt{H_d/h}} + \mathbf{M})\mathbf{V} \tag{15}$$

where the values in mask $\mathbf{M}$ are shown by black and white rectangles in Fig. 9. $h = 12$ denotes the number of heads. $\mathbf{Z}_i^T(t) = [\mathbf{Z}_{i,1}^T(t), ..., \mathbf{Z}_{i,N_q+S_l}^T(t)]^T \in \mathbb{R}^{(N_q+S_l) \times H_d}$. Only the last token $\mathbf{Z}_{i,N_q+S_l}^T(t)$ is fed into feed-forward layer because the last token is informative about the complete

Table 4: Performance (AUC, %) on the benchmarks. ST, Ave, UB, Ped2 and NWPU represent ShanghaiTech, CUHK Avenue, Ubnormal, UCSD Ped2 and NWPU Campus, respectively. Macro-AUC and micro-AUC Reiss & Hoshen (2022) are evaluated.

| Algorithm | ST | Ave | UB | Ped2 | NWPU |
|---|---|---|---|---|---|
| Ours with VLM based detector | 92.7 / 88.9 | 94.9 / 94.5 | 86.8 / 81.5 | 99.8 / 99.1 | 73.5 / 71.6 |
| Ours with VLM based detector (w/o box expansion) | 92.3 / 88.4 | 93.6 / 93.2 | 86.6 / 81.2 | 99.7 / 99.0 | 73.0 / 71.2 |
| Ours with Yolo detector Wang et al. (2023) | 92.8 / 89.0 | 94.9 / 94.5 | 86.9 / 81.5 | 99.8 / 99.1 | 73.7 / 71.7 |
| Ours with Yolo detector (w/o box expansion) | 92.3 / 88.5 | 93.7 / 93.3 | 86.6 / 81.3 | 99.7 / 99.0 | 73.0 / 71.2 |
| Sliding windows | 80.8 / 77.6 | 80.5 / 79.9 | 75.4 / 72.9 | 88.6 / 87.8 | 66.5 / 64.4 |
| Ours with VLM based detector, RM with 5 layers | 91.3 / 87.4 | 92.8 / 92.1 | 85.1 / 80.4 | 99.0 / 98.6 | 72.0 / 70.1 |
| Ours with VLM based detector, RM with 9 layers | 91.4 / 87.3 | 92.9 / 92.3 | 85.4 / 80.6 | 99.4 / 98.9 | 72.2 / 70.3 |

sequence. The feed-forward layer has $H_d$ input channels and $N_{vocabulary}$ output channels, producing $N_{vocabulary} = 30,523$ probabilities indicating the likelihood of candidate tokens. $N_{vocabulary}$ is vocabulary size, according to BERT tokenizer Devlin et al. (2018). The best candidate $\mathbf{S}_{i,k+1}(t)$ is appended to the end of sequence $\mathbf{S}_{i,1}(t), \mathbf{S}_{i,2}(t), ..., \mathbf{S}_{i,k}(t)$ before beginning the next iteration. The iterations terminate upon generating the whole sequence $\mathbf{S}_i(t)$ with length $S_l$. This module is trained with cross-entropy loss.

## D  ABLATION STUDY ON THE METHOD FOR OBJECT DETECTION

Table 4 shows the comparison between using VLM Bai et al. (2023), YOLOWang et al. (2023) and sliding windows for object detection. Specifically, window sizes are fixed as follows: 224 for ShanghaiTech, 320 for CUHK Avenue, 320 for Ubnormal, 60 for UCSD Ped2, and 224 for NWPU Campus, with the aim of including largest objects. The results indicate that effective object detection is crucial for accurate performance. Furthermore, the comparisons between the settings with and without bounding box expansions show that bounding box expansions contribute to capturing more contextual information, benefiting performance.

## E  ABLATION STUDY ON ITAM'S STRUCTURE

Table 5 shows the influences of ITAM's structures and training data on performance. Setting 1 is the default setting with "Str. 1" and "D. 1". "Str. 1" denotes the structure Li et al. (2023) shown in Section 3.2 and "D. 1" denotes the training data of BLIP-2 Li et al. (2023). In "Str. 1", the image transformer for feature extraction has 6 transformer layers each of which includes one self-attention layer, one cross-attention layer and one feed-forward layer. Both of the self-attention layer and the cross-attention layer have 12 heads. In "Str. 2", the numbers of heads are changed to 6 with other settings fixed. In "Str. 3", the number of sequential transformer layers is changed to 3 with other hyperparameters unchanged. "D. 2" refers to the configuration where ITAM is trained on the training set of anomaly detection benchmark in each experiment. These training sets include only normal events. The captioning labels on benchmarks' training data are generated by running the pre-trained BLIP-2 model Li et al. (2023) on the normal videos. It can be seen that the structure and data variations do not significantly influence performance as long as image-text alignment is conducted. More importantly, ITAM can be trained using normal data and detect unseen anomalies.

Setting 5 and 6 show that the number of image tokens $N_q$ does not significantly influence performance. Setting 7 shows that if SMM is trained using the captioning labels from dataset Lin et al. (2014) and without requiring Qwen-Chat, performance is not influenced. For instance, if the captioning label of an image is "The man is running" which prompts SMM to output "yes", then we

Table 5: Ablations of ITAM's structure using Micro-AUC. TME and TSG are short for Temporal Motion Estimation and Temporal Sentence Generation, respectively. $N_q$ is the number of image tokens.

| Setting | Stage 1 | Stage 2 | $N_q$ | TME | TSG | $N_{SMM}$ | ST | Ave | UB | Ped2 |
|---------|---------|---------|-------|-----|-----|-----------|------|------|------|------|
| 1 | Str. 1, D. 1 | ✓ | 32 | ✓ | ✓ | 3 | 88.9 | 94.5 | 81.5 | 99.1 |
| 2 | Str. 1, D. 2 | ✓ | 32 | ✓ | ✓ | 3 | 88.9 | 94.5 | 81.5 | 99.1 |
| 3 | Str. 2, D. 1 | ✓ | 32 | ✓ | ✓ | 3 | 88.6 | 94.1 | 81.3 | 99.1 |
| 4 | Str. 3, D. 1 | ✓ | 32 | ✓ | ✓ | 3 | 88.7 | 94.4 | 81.2 | 99.1 |
| 5 | Str. 1, D. 1 | ✓ | 64 | ✓ | ✓ | 3 | 88.8 | 94.5 | 81.5 | 99.1 |
| 6 | Str. 1, D. 1 | ✓ | 128 | ✓ | ✓ | 3 | 88.9 | 94.6 | 81.5 | 99.1 |
| 7 | Str. 1, D. 1 | ✓ | 32 | ✓ | SMM w/o Qwen | 3 | 88.9 | 94.5 | 81.5 | 99.1 |

randomly sample another sentence with a different meaning, such as "The man is fighting", which causes SMM to output "no". Implementations are based on NLTK library Hardeniya et al. (2016).

# F    ABLATION STUDY ON THE NUMBER OF LAYERS IN RM

Table 4 compares the performance of our RM with 7 layers to configurations with 5 layers and 9 layers, respectively. It can be seen that 7 is a better choice.

# G    PROCEDURES FOR GENERATING QUESTIONS IN TSGM

As is shown in Fig. 2(b), TSGM firstly converts the declarative sentence "The man is pushing a stroller on the street." to an interrogative sentence "Is the man pushing a strollor on the street?" The conversion is based on nltk library Hardeniya et al. (2016) and the procedures are shown in Algorithm 2:

---

**Algorithm 2** Algorithm for Converting Declarative Sentences to Interrogative Sentences

---

1: Input sentence: $\mathbf{D} \leftarrow \mathbf{S}_i^I(t-1) = $ 'The man is pushing a stroller on the street.'
2: Tokenization: $\mathbf{D} \rightarrow$ ['The', 'man', 'is', 'pushing', 'a', 'stroller', 'on', 'the', 'street', '.']
3: Locate first verb: $\mathbf{D}_{firstverb}$ = 'is'
4: Divide sentence using first verb: $\mathbf{D} \rightarrow \mathbf{D}_1 + \mathbf{D}_{firstverb} + \mathbf{D}_2$, $\mathbf{D}_1$ = 'The man', $\mathbf{D}_2$ = 'pushing a stroller on the street'
5: Change the order of parts: $\mathbf{Q} \leftarrow \mathbf{D}_{firstverb} + \mathbf{D}_1 + \mathbf{D}_2$
6: **return  Q**

---

# H    OPERATIONAL EFFICIENCY

All experiments are conducted on an NVIDIA A100 GPU and an Intel(R) Xeon(R) Gold 6248R CPU. For object detection, we have employed both YOLOv7 detector Wang et al. (2023) and another detector based on Qwen-VL-7B Bai et al. (2023). As is shown by Table 4, both detectors achieve similar accuracy. In terms of inference speed, the YOLO detector Wang et al. (2023) processes each frame in 1.5 milliseconds, whereas Qwen-VL-7B Bai et al. (2023) requires 5.2 seconds per frame. Consequently, we evaluate the operational efficiency of the proposed framework using the YOLO detector. The inference times of all components in the proposed framework are measured and summarized in Table 6. With all components considered, the proposed method achieves an average frame rate of 12 FPS with an average of 5 objects per frame. The average number of 5 objects is based on the findings of Wang et al. (2022a).

In the future, we aim to explore the methods that utilize a fixed number of bounding boxes per frame to maintain a constant inference time, even with an increased number of objects. Additionally, we

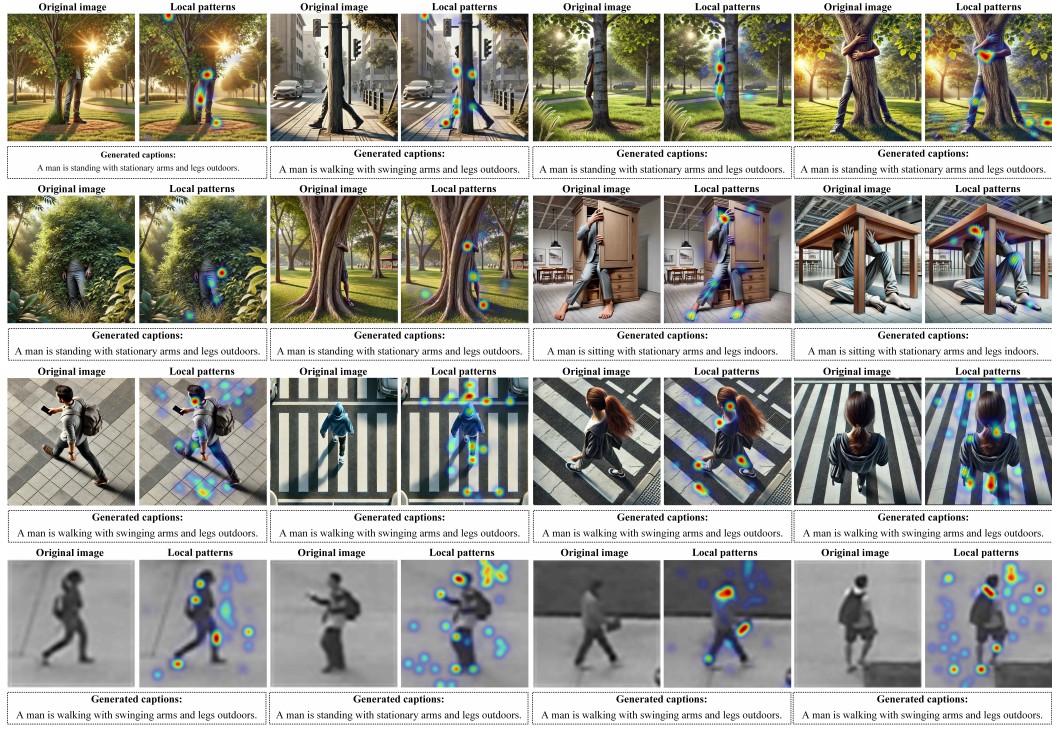

Figure 10: Visualization of the heatmaps of local patterns under occlusions, viewpoint changes and low resolutions.

will investigate parsing multiple objects within a single bounding box to maintain a fixed number of bounding boxes per frame.

Table 6: Runtime and memory consumption of different modules in the proposed framework, runtime is measured in milliseconds(ms). Inference is conducted with batch size 256.

| Modules | Object detection (YOLO) | Backbone and ITAM | TSGM (SMM) | TSGM (IGTG) | CMAM | TME | RM |
|---|---|---|---|---|---|---|---|
| Runtime | 2.5 | 10.1 | 1.5 | 5.9 | 0.0078 | 2.6 | 0.28 |
| GPU Memory (Gigabytes) | 2.78 | 18.88 | 0.63 | 13.54 | 0.04 | 0.0 | 0.55 |

Table 7 compares the proposed approach with baseline LLM-based AnomalyRuler Yang et al. (2024a). AnomalyRuler involves a VLM Processing stage with CogVLM-17B and a LLM Reasoning stage with GPT-4, consuming 192.56 ms and 504.79 ms per frame on NVIDIA A100 GPU, respectively .

Table 7: Comparison between the proposed method and LLM-based anomaly detector Yang et al. (2024a). Runtime measured in milliseconds(ms), performance measured in AUC (%).

| Methods | Runtime per frame | Performance on Shanghaitech | Performance on Avenue |
|---|---|---|---|
| Ours | 83.95 | 88.9 | 94.5 |
| AnomalyRuler Yang et al. (2024a) | 697.35 | 85.2 | 89.7 |

# I    VISUALIZATION OF LOCAL PATTERNS UNDER OCCLUSIONS AND VIEWPOINT CHANGES

In real-world surveillance videos, occlusions, viewpoint variations and low-resolution conditions are common. Fig. 10 shows some examples of the local patterns identified by image-text alignment and cross-modality attention. The local patterns capture semantically meaningful features such as body joints which are consistent across the variations. The compact representations ignore redundant details and contribute to generalizable embeddings.

# J    FUTURE WORK

One limitation of the current framework is the reliance on object detectors. Currently, the performance of current Vision-Language Models (VLMs) is limited by their fields of view. For example, when processing an image with a large scene, a vision-language model tends to overlook many details, highlighting the necessity of object detectors that facilitate the processing of local regions independently. Table 4 shows that object detectors significantly outperform sliding windows, the poor performance of the latter may result from an incorrect strategy. As a result, we will try more efficient and effective ways to parse events in complex scenes and images with large fields of view where many objects reside. Specifically, we will explore the integration of object detectors in an end-to-end large model. In simpler scenes with fewer objects, an input image is embedded with fewer vision tokens. As scenes become more complex, more objects are involved, then an input image is encoded with an increased number of vision tokens each of which describes one or more objects. Besides, we will explore ways to improve efficiency.

