# OpenReview forum: "Local Patterns Generalize Better for Novel Anomalies"
_ICLR.cc/2025/Conference — ICLR 2025 Poster_

### Official Review · Reviewer_mbj7 · 2024-10-15

**Soundness:** 4
**Presentation:** 4
**Contribution:** 3
**Rating:** 6
**Confidence:** 3

**Summary:**

This article is about video anomaly detection (VAD). This paper proposes a framework for recognizing local patterns, which can be generalized to new samples and dynamic modeling of local patterns. This paper proposes image-text alignment and cross-modal attention. Generalizable representations are built by focusing on textual information features that filter out unnecessary differences in visual data. In addition, time motion estimation complements spatial local models to detect anomalies characterized by new spatial distributions or unique dynamics. A large number of experiments have verified the effectiveness.

**Strengths:**

1. The two-stage training method of this paper is reasonable. And allow for more fine-grained local features.
2. This paper gives a lot of visualizations to make it easier to understand the specific content.
3. The paper has achieved good performance, and the ablation experiment is given.

**Weaknesses:**

1. Not giving motivation for each part of the method. In my opinion, a good paper should give a specific reason and then introduce the method.
2. The efficiency of the model is worth discussing. You have proposed a lot of model modules. How much more reasoning time will they add to the network?

**Questions:**

My understanding of this field is unprofessional. So I will further follow the opinions of other reviewers.

---

> ### Author Response · Authors · 2024-11-24
> **Explanations and revisions**
>
> Thank you for your constructive comments which have helped us greatly in revising our manuscript.
>
> 1. Motivation for each part of the method
>
> Firstly, we have clarified the purpose of the framework in the refined Fig. 1 in rebuttal revision. Global patterns retain redundant visual details, whereas local patterns, learned via image-text alignment, capture semantically meaningful components such as body joints. This alignment with textual descriptions enables local patterns to focus on relevant semantic elements, facilitating the comparison of spatial distributions of local patterns across images for identifying anomalies.
> Secondly, we have clarified the purpose of the proposed two-stage scheme, including ITAM and CMAM, in Section 3.2 and 3.3. Image-text alignment in Stage 1 facilitates the identification of local components. When encountering an unseen action, such as running, the model can recombine known components like arms and legs to generate descriptive language that captures the essence of the action without explicitly naming it. As illustrated in Fig. 1. The generalizable components are shared by normal and novel abnormal events.
> We have added experiments in “Section 4.3 Ablation Studies” and Appendix E with ablations on ITAM’s structure. As is illustrated in Setting 1, 2, 3 and 4 in Table 5, the structure and training data for ITAM do not significantly influence performance. The core contribution is image-text alignment. We have also enriched the ablations on CMAM in Setting 3, 4 and 5 in Table 3. It can be seen that cross-modality feature fusion outperforms either single modality even though the texts are generated from image tokens. In Setting 2 of Table 5, ITAM is trained using only normal samples and achieves good performance. As a result, image-text alignment is capable of generalizing to novel samples.
> Thirdly, we have clarified the motivation of SMM in TSGM in Section 3.4. In some cases, a person walks from near to far, causing the resolution of the person to gradually decrease and leading to inaccurate captions. Therefore, SMM in TSGM plays a vital role in determining whether earlier events captured in high-resolution moments are still represented by the later image tokens, it captures inter-frame dependencies and refines sentence coherence. Setting 6 and 7 in Table 3 shows the benefits of SMM. Section 4.3 also ablates on the structure of SMM.
> The abstract and introduction are also updated accordingly.
>
> 2. The efficiency of the model is worth discussing
>
> We have added Appendix H with Table 6 and Table 7 for showing operational efficiency, including the runtime of each module and spatio-temporal complexity. In comparison with baseline LLM-based approaches, the proposed approach shows advantages in both accuracy and speed.
> We have uploaded the code for implementing the proposed approach in the url at the end of Abstract. Currently, more details as well as hand-on manuals are provided, including the Backbone, ITAM, CMAM, TSGM, SMM and so on.
>
> 3. Additional information
>
> The core contributions of this paper come in the following ways:
> (1)	An approach is proposed to represent novel anomalies using local patterns. Local patterns capture semantically meaningful components such as body joints and are consistent across domains and generalize well, reducing the redundant visual details in global patterns. We have refined Fig. 1, the first paragraphs in Section 3.2, 3.3, 3.4 and added Appendix I to illustrate this motivation.
> (2)	A two-stage process with image-text alignment and cross-modality attention is proposed for identifying the local patterns. The ablation studies on ITAM’s structure and training data are detailed in Appendix E. ITAM can be trained using normal data and detect unseen anomalies. The primary contributor to generalization is image-text alignment instead of external pre-trained models. Besides, the texts are generated from image tokens in the proposed scheme for cross-modality fusion.
> (3)	The generation of captions is influenced by visual data variances such as low resolutions. As is shown in Fig. 2(b) and Section 3.4. Therefore, SMM determines whether earlier high-resolution events are represented by later image tokens. It captures inter-frame dependencies and refines sentence coherence. SMM uses earlier text tokens to generate precise captions for low-resolution observations. Ablation studies in Section 4.3 show that the proposed SMM achieves this goal by stacking state machines, outperforming single state machines.

---

> > ### Comment · Reviewer_mbj7 · 2024-11-24
> >
> > Thanks for your reply and hard work, you have solved my doubts, and the paper has become more substantial. Considering the innovation and contribution to the field, I think this paper still cannot be raised to 8 points, and it is worthy of 6 points.

---

### Official Review · Reviewer_qaRU · 2024-10-30

**Soundness:** 3
**Presentation:** 2
**Contribution:** 3
**Rating:** 5
**Confidence:** 4

**Summary:**

This paper introduces a novel framework for video anomaly detection (VAD) that prioritizes identifying local patterns over conventional global patterns. The authors contend that local patterns generalize better to novel anomalies that were not encountered during training. Their proposed approach follows a two-stage process involving image-text alignment and cross-modality attention to efficiently capture and model local patterns. Additionally, the framework includes a State Machine Module (SMM) to integrate temporal dynamics, enabling enhanced anomaly detection by leveraging both spatial and temporal cues. Experimental results show that this approach achieves state-of-the-art performance on well-established benchmark datasets.

**Strengths:**

1. The proposed framework is well-structured, with thoughtfully implemented methods.
2. Experimental results confirm that the approach achieves state-of-the-art performance on established benchmark datasets for video anomaly detection.
3. The method focuses on fine-grained anomaly features and employs text-image alignment to effectively capture local patterns.
4. It incorporates Long-range Memory technology, specifically HiPPO, into video anomaly detection.

**Weaknesses:**

1. The core of the proposed SMM module comes from works like HiPPO, so it seems that the proposed SMM is directly applying these modules to the VAD task.
2. Both the Image-Text Alignment Module and Cross-Modality Attention Module are based on pre-existing techniques, which limits the methodological innovation.
3. Is the observed performance improvement attributed to the additional large vision-language models, such as Qwen-VL and BLIP2? The comparison may not be entirely fair. It would be beneficial if the authors could provide evidence or experimental results to clarify whether these powerful external models are the primary contributors to the performance gains.
4. The motivation for the work lacks clarity. How do the image-text alignment and cross-modality attention modules achieve “identification of local patterns that are consistent across domains and generalize well”? Additionally, how do they contribute to “generalizing model representations to novel anomalies”?
5. Certain claims may require further validation, such as the statement: “the complementary relation between visual and textual features remains underexplored.”
6. The paper lacks runtime and efficiency analysis. The code introduction is incomplete, and several experimental details are missing, such as the specific version and scale of Qwen-VL used.

**Questions:**

1. The Global and Local Pattern representations in Figure 1 are hand-drawn, which limits their reliability. Are there any real feature visualization images available instead? Using actual visualizations could better illustrate the motivation and effectiveness of the proposed method, particularly in showing whether it yields more distinguishable local patterns. Figure 1 alone does not provide enough information to convey the method’s motivation and impact.
2. Utilizing Qwen for cropping bounding box regions based on prompts could significantly impact efficiency.
3. Is the introduction of the Qwen-Chat model the primary source of performance improvement? My concern is that the proposed method incorporates numerous external models, and it remains unclear whether these additions are the main contributors to the observed performance gains.
4. Could smaller models be used to replace these large multimodal models? If so, would this result in a significant decrease in performance?
5. Could you provide statistical results on runtime and efficiency? Does the proposed method have a significant impact on operational efficiency?

---

> ### Author Response · Authors · 2024-11-24
> **Explanations and Revisions**
>
> Thank you for your comments.
>
> 1. Difference from HiPPO
>
> The purpose of SMM is to address the influences of low-resolution conditions that hinder precise captioning, as is shown by Section 3.4, Fig. 2(b) and Fig. 3 in rebuttal revision. SMM determines whether earlier high-resolution events are represented by later image tokens. We have provided the code for implementing SMM in the anonymous url, as well as hand-on manuals. It can be seen that SMM significantly differs from HiPPO.
> Different from HiPPO which tackles single-modality 1-dimensional signals, the proposed SMM models the complex dependencies in high-dimensional multi-modal sequences by stacking up 3 state machines. The advantages of stacking up state machines is shown by ablation studies in Section 4.3 and Table 3.
> It can be seen that the SMM with stacked state machines outperforms a single state machine.
>
> 2. Is the observed performance improvement attributed to the additional large vision-language models
>
> We have enriched Section 4.3 and Appendix E with ablations on the structure of Image-Text Alignment Module (ITAM) in the rebuttal revision. Specifically, we have made significant changes to the structure and training data of ITAM without much influence on performance. It can be seen from Table 5 that the structure and data variations does not significantly influence performance as long as image-text alignment is conducted. ITAM can be trained using normal data and detect unseen anomalies. As a result, the core contribution is image-text alignment instead of pre-existing models.
> In terms of CMAM, the comparison between Setting 3, 4 and 5 in Table 3 illustrates the benefits of both modalities. The texts are generated from image tokens in our proposed cross-modality fusion.
> As is addressed in Section 3.1, Appendix D (Table 4), the YOLO detector contributes to the same accuracy as Qwen-VL based detector.
>
> 3. The motivation for the work lacks clarity
>
> We have augmented Fig. 1, Section 3.2, 3.3 and 3.4 with clear motivations. Texts describe generic movement attributes (e.g., "A man is walking with swinging arms and moving legs"). When encountering an unseen action, such as running, the model can recombine known components like arms and legs to generate descriptive language that captures the essence of the action without explicitly naming it. As illustrated in Fig. 1 and Fig. 10.
>
> 4. Certain claims may require further validation
>
> The claim has been removed. In Stage 2 of the framework, image tokens and text tokens are combined in CMAM. Table 3 now includes Setting 3, 4 and 5 to highlight the necessity of both modalities. Besides, Fig. 5 shows that combing visual and textual features outperforms using either modality alone.
>
> 5. Runtime and efficiency analysis
>
> We tried both Qwen-VL-7B and YOLO-v7 as options for object detection. As is shown by Table 4 in Appendix D, both detectors achieve similar accuracy. Appendix H, Table 6 and Table 7 are added to detail the inference times of all components in the framework. With all components considered, the method achieves an average frame rate of 12 FPS.
>
> 6. The Global and Local Pattern representations in Figure 1 are hand-drawn
>
> We have carefully revised Fig. 1 in the rebuttal revision. Currently, global and local patterns are visualized using real feature heatmaps. As shown in Fig. 1, local patterns capture semantically meaningful components such as body joints.
>
> 7. Smaller models replace large ones, Qwen has low efficiency
>
> We tried both Qwen-VL-7B and YOLO-v7 as options for object detection. As is shown by Table 4 and 6 in the Appendix, both detectors achieve similar accuracy. YOLO is faster than Qwen-VL-7B.
> In terms of BLIP-2, we have enriched Section 4.3 and added Appendix E with ablations on the structure of ITAM. Table 5 shows that the structure and data variations does not significantly influence performance. ITAM can learn from normal data and detect anomalies.
>
> 8. Primary source of performance improvement
>
> We have enriched Section 4.3 and added Appendix E for ablation studies. Our first contribution is the two-stage scheme for identifying local patterns. The first stage is image-text alignment with motivation being clarified in the revised Section 3.2. Appendix E shows that the structure and training data of ITAM do not significantly influence model performance. The contribution of the second stage is presented by the ablations on Cross-Modality Attention.
> The second contribution is SMM which combines the image features from current moment with the text features from the previous moment in augmenting the descriptions about images. The ablation studies on SMM’s structure are detailed in Section 4.3.
> Qwen model is only leveraged when generating the training labels for SMM, it does not influence inference speed, according to Section 3.4. Setting 7 in Table 5 shows that if SMM is trained using the captioning labels from dataset and without requiring Qwen-Chat, performance is not influenced.

---

### Official Review · Reviewer_XFDv · 2024-10-30

**Soundness:** 3
**Presentation:** 3
**Contribution:** 3
**Rating:** 6
**Confidence:** 3

**Summary:**

This paper proposes a novel framework for video anomaly detection (VAD) that focuses on identifying local patterns to better generalize to unseen anomalies.  The framework employs a two-stage process: first, it uses image-text alignment to locate local patterns that are consistent across visual data variances; second, it refines these patterns using cross-modality attention.  To further enhance the model, the authors introduce temporal clues through a State Machine Module (SMM) and temporal motion estimation.

**Strengths:**

The proposed two-stage framework for identifying local patterns is novel and well-motivated. The use of image-text alignment and cross-modality attention is interesting and potentially useful.

**Weaknesses:**

1) The paper lacks a clear discussion of the computational complexity of the proposed framework. Given the use of large language models (LLMs) and other complex modules, it is important to address the efficiency of the approach.
2) What's the role of State Machine Module (SMM) in temporal sentence generation, there need more detailed explanation of the SMM and its role.
3) How does the proposed method handle situations with significant occlusions or viewpoint changes, which are common in real-world surveillance videos?
4) The two-stage process for extracting spatial local patterns using image-text alignment and cross-modality attention is not explained in enough detail. The paper lacks a clear, step-by-step explanation of how these complex processes work.

**Questions:**

The paper mentions limitations related to the reliance on VLM-based object detectors. How can this limitation be addressed in future work?

---

> ### Author Response · Authors · 2024-11-24
> **Explanations and revisions**
>
> Thank you for your constructive comments which have helped us greatly in revising our manuscript.
>
> 1. Clear discussion of the computational complexity
>
> Appendix H, Table 6 and Table 7 are added in rebuttal revision to detail the inference times of all components in the proposed framework. With all components considered, the proposed method achieves an average frame rate of 12 FPS, achieving advantages in both accuracy and speed.
> We have uploaded the code for implementing the proposed approach in the url at the end of Abstract. Currently, more details as well as hand-on manuals are provided, including the Backbone, ITAM, CMAM, TSGM, SMM and so on.
> We tried both Qwen-VL-7B and YOLO-v7 as options for object detection. As is shown by Table 4 in Appendix D, both detectors achieve similar accuracy. In terms of inference speed, the YOLO detector processes each frame in 1.5 milliseconds, whereas Qwen-VL-7B requires 5.2 seconds per frame. Consequently, we evaluate the operational efficiency of the proposed framework using the YOLO detector.
>
> 2. Role of State Machine Module (SMM) in temporal sentence generation
>
> We have carefully revised Fig. 2(b), Fig. 3 and Section 3.4 to detail SMM and its role. Stage 2 of the framework addresses the visual data variances, such as low resolutions, that influence the generation of captions. As is shown in Fig. 2(b), the module for image-grounded text generation only provides a coarse caption "A man is walking" on later observations because of low-resolutions, it is not as precise as earlier captions "A man is pushing a stroller on the street" even if they actually describe the same event.
> Therefore, SMM in TSGM plays a vital role in determining whether earlier events are still represented by the later image tokens, it captures inter-frame dependencies and refines sentence coherence.
> Specifically, SMM takes in the concatenation of image tokens at t with the text tokens of the sentence generated at t-1, it determines whether the event described by the sentence still resides in the image tokens, and returns “yes” or “no”.
> In terms of SMM’s structure, we have added ablation studies on SMM’s structure in Section 4.3 of the rebuttal revision. We have uploaded the code for training and inference with SMM in the url at the end of Abstract.
>
> 3. Handle situations with significant occlusions or viewpoint changes
>
> Firstly, Fig. 10 is added to Appendix I of rebuttal revision, the figure shows some examples of the local patterns identified by image-text alignment and cross-modality attention. The local patterns capture semantically meaningful features such as body joints which are consistent across the variations. The compact representations ignore redundant details and contribute to generalizable embeddings.
>
> 4. Explanations about two-stage process for extracting spatial local patterns
>
> Firstly, we have enriched Section 3.2 in rebuttal revision for better illustration. Algorithm 1 is added in Section 3.2 to detail the workflow. Stage 1 identifies the image tokens using image-text alignment, Stage 2 further refines and captures local patterns using cross-modality attention. The processes identify the semantically meaningful components in each image region that align with texts. Specifically, the captions used to train the model describe generic movement attributes (e.g., "A man is walking with swinging arms and moving legs"). When encountering an unseen action, such as running, the model can recombine known components like arms and legs to generate descriptive language that captures the essence of the action without explicitly naming it. As illustrated in Fig. 1.
>
> 5. Limitations related to the reliance on VLM-based object detectors
>
> We tried both Qwen-VL-7B and YOLO-v7 as options for object detection. As is shown by Table 4 in Appendix D, both detectors achieve similar accuracy. In terms of inference speed, the YOLO detector processes each frame in 1.5 milliseconds, whereas Qwen-VL-7B requires 5.2 seconds per frame. Consequently, we use the YOLO detector in the framework.
> In the future, we will explore the integration of object detectors in an end-to-end large model. In simpler scenes with few objects, the whole input image is embedded with fewer vision tokens. As the scenes become more complex, more objects are involved, then the input image is encoded with an increased number of vision tokens each of which describes one or more objects.
> This has been added to Appendix J.

---

### Official Review · Reviewer_Yam7 · 2024-11-03

**Soundness:** 3
**Presentation:** 4
**Contribution:** 3
**Rating:** 6
**Confidence:** 3

**Summary:**

The paper introduces a novel framework for video anomaly detection, aiming to improve generalization for detecting new, unseen anomalies by focusing on local patterns rather than global event patterns. Traditional video anomaly detection (VAD) methods often struggle with unseen anomalies, as they primarily analyze global patterns. This framework utilizes image-text alignment and cross-modality attention to identify and refine local patterns while enhancing them with temporal information. Core components include the Image-Text Alignment Module (ITAM), Cross-Modality Attention Module (CMAM), and State Machine Module (SMM). The proposed approach demonstrates superior performance on several benchmark datasets, suggesting it can generalize better to novel anomalies.

**Strengths:**

* By using image-text alignment and cross-modality attention, this method successfully extracts local patterns that remain consistent across varying visual data, enhancing its ability to detect novel anomalies.

* The State Machine Module (SMM) and motion estimation integrate temporal clues, effectively strengthening the detection capabilities by including sequential information for more accurate anomaly detection.

*By combining visual and textual features in identifying local patterns, the model benefits from enhanced robustness and accuracy across different visual domains.

**Weaknesses:**

* The method relies on the detection effect of visual-linguistic modeling (VLM), whereas multi-object image processing may ignore contextual information and affect performance. The authors need to provide more analysis on the ablation of the foundational models.

* The need for multiple layers of modules (e.g., ITAM, CMAM, SMM) to work jointly results in a complex training process that consumes more time and resources. Please provide a comparison of the spatio-temporal complexity analysis with previous methods to demonstrate the practical effectiveness of the method.

* In low-resolution scenes, the generated text description loses detail information, which affects the anomaly detection effect.

**Questions:**

* How to further improve the generalization of local patterns without relying on visual-linguistic models?

* How does the method ensure adaptability to low-resolution videos in different datasets and real-world application scenarios?

---

> ### Author Response · Authors · 2024-11-24
> **Explanations and revisions**
>
> Thank you for your constructive comments which have helped us greatly in revising our manuscript.
>
> 1. More analysis on the ablation of the foundational models. Multi-object image processing may ignore contextual information and affect performance
>
> As is shown by the implementation details in Section 4.1 in rebuttal revision, we have expanded each bounding box horizontally and vertically by 50% on both sides. The benefits of box expansion are shown in Table 4 of Appendix D, the comparisons between the settings with and without bounding box expansions show that bounding box expansions contribute to capturing more contextual information.
>
> 2. Comparison of the spatio-temporal complexity analysis with previous methods
>
> We have added Appendix H in the rebuttal revision for showing operational efficiency. In comparison with the baseline LLM-based approach, the proposed approach shows advantages in both accuracy and speed.
> We have uploaded the code for implementing the proposed approach in the url at the end of Abstract. Currently, more details as well as hand-on manuals are provided, including the Backbone, ITAM, CMAM, TSGM, SMM and so on.
>
> 3. Adaptability to low-resolution videos
>
> Firstly, we have added Fig. 10 to Appendix I of rebuttal revision, visualizing the local patterns and generated captions under low resolutions, occlusions and viewpoint changes. It can be seen that TSGM generates accurate captions and captures semantically meaningful features such as body joints, in a similar fashion as in high-resolution images.
> Secondly, we have clarified the purpose of SMM in Section 3.4. In some cases, a person walks from near to far, causing the resolution of the person to gradually decrease, leading to inaccurate captions. Therefore, SMM in TSGM plays a vital role in determining whether earlier events captured in high-resolution moments are still represented by the later image tokens, it captures inter-frame dependencies and refines sentence coherence.
>
> 4. Further improve the generalization of local patterns without relying on visual-linguistic models
>
> Firstly, we have clarified the purpose in Fig. 1. As shown in Fig. 1, global patterns retain redundant visual details, whereas local patterns, learned via image-text alignment, capture semantically meaningful components such as body joints. This alignment with textual descriptions enables visual local patterns to focus on relevant semantic elements, facilitating the comparison of spatial distributions of local patterns across images for identifying anomalies.
> Secondly, we have clarified the purpose of the proposed two-stage scheme in Section 3.2. The model learns to decompose global patterns into semantically meaningful local patterns. When encountering an unseen action, such as running, the model can recombine known components like arms and legs to generate descriptions that capture the essence of the action without explicitly naming it.
> As a result, our motivation is to replace global patterns which include redundant details with semantic local patterns which correspond to generalizable components. Normal and novel abnormal events share the components. If without visual-linguistic models, we can try another way to identify the components as local patterns, such as using graph representations.

---

### Author Response · Authors · 2024-11-25
**Explanations and revisions**

Dear Reviewers, thank you for your initial feedback on our submission. We have addressed your comments and provided detailed responses in our rebuttal. Please let us know if there are any additional points you’d like us to clarify before the discussion phase concludes. Your feedback is highly appreciated.

---

### Author Response · Authors · 2024-11-26
**Further explanations and revisions**

Dear everyone, thank you for your initial feedback on our submission. We have further addressed the comments and refined the responses in our rebuttal, also we have uploaded the revised paper & code. Please let us know if there are any additional points you’d like us to clarify before the discussion phase concludes. Your feedback is highly appreciated.

---

### Author Response · Authors · 2024-12-01
**Clarification on concerns**

Dear Reviewers and Area Chairs,

I would like to express my gratitude for your time and feedback. We have carefully addressed the concerns raised by the reviewers in our previous responses.

In terms of novely, the structure of the proposed SMM significantly differs from HiPPO. We have provided the code in our submission, which clearly demonstrates the differences between our approach and the referenced work.  Besides, we have innovatively discovered the local patterns that can be recombined to capture the essence of novel anomalies, this is implemented by the two-stage process with Image-Text Alignment and Cross-Modality Attention to gradually identifying the generalizable local patterns. Ablation studies show that our method also works with smaller models and do not necessarily rely on pre-existing large models.

In terms of efficiency concerns, we have provided spatio-temporal complexity analysis and comparisons with previous methods, and we have two choices of object detectors one of which has a much higher efficiency than VLM. The efficiency of the approach exceeds baseline LLM-based pproach.

We have clarified the motivations and implementations of image-text alignment, cross-modality attention and temporal sentence generation with stacked state machines.

In cases of low-resolution scenes, occlusions or viewpoint changes, our proposed local patterns capture semantically meaningful features such as body joints which are consistent across the variations. The proposed SMM determines whether earlier high-resolution events are represented by later image tokens, it leverages inter-frame dependencies to deal with variations. In the concerns about the ignorance of contextual information, we have actually expanded bounding boxes and facilitating the involvement of relative contexts in analyzing subjects, benefiting performance.

We would greatly appreciate any further feedback or clarification requests from the reviewers. If there are any additional questions, we are happy to provide further details.

Thank you once again for your attention and consideration.

Best regards

---

### Meta-Review · Area_Chair_hNts · 2024-12-20

**Metareview:**

# Summary and Recommendation for Acceptance

---

## **Strengths**
1. **Novel Contributions**:
   - Proposes a novel framework for video anomaly detection (VAD) focusing on **local patterns** rather than global patterns, which improves generalization to unseen anomalies.
   - Introduces a **two-stage process**:
     - **ITAM**: Captures semantically meaningful local components.
     - **CMAM**: Refines local patterns using cross-modality fusion.
   - Enhances temporal reasoning with a **State Machine Module (SMM)**, addressing low-resolution scenarios by leveraging inter-frame dependencies.

2. **Robust Experimental Validation**:
   - Achieves state-of-the-art performance on well-established benchmark datasets.
   - Ablation studies demonstrate the effectiveness of individual components, including ITAM, CMAM, and SMM.

3. **Generalization Capabilities**:
   - Local patterns remain robust across varying visual conditions (e.g., low resolution, occlusion, and viewpoint changes).
   - The model recombines known components (e.g., arms, legs) to describe unseen actions without explicitly naming them.

4. **Efficiency**:
   - Offers competitive spatio-temporal complexity, achieving an average frame rate of 12 FPS, outperforming baseline methods in both speed and accuracy.

5. **Community Contribution**:
   - Code and detailed manuals provided to ensure reproducibility and ease of adoption.

---

## **Weaknesses**
1. **Complexity**:
   - The framework includes multiple components (ITAM, CMAM, SMM) that require careful integration, making training and implementation challenging.

2. **Heavy Reliance on Pre-trained Models**:
   - Use of vision-language models (e.g., Qwen-VL-7B) raises concerns about whether performance improvements are mainly due to these external models.

3. **Clarity and Presentation**:
   - Initial version lacked clear motivation and step-by-step explanations for components like ITAM and CMAM.
   - Fig. 1 in the initial submission was hand-drawn, reducing its effectiveness in illustrating local and global pattern distinctions.

4. **Efficiency Analysis**:
   - Earlier versions did not provide detailed runtime and efficiency analyses for individual modules.

---

## **Authors' Mitigation**
1. **Clarity and Motivation**:
   - Revised the manuscript to provide step-by-step explanations for ITAM, CMAM, and SMM.
   - Updated Fig. 1 with real feature heatmaps to visually distinguish local and global patterns.
   - Added Algorithm 1 in Section 3.2 to detail the workflow of the two-stage process.

2. **Ablation Studies**:
   - Conducted extensive ablation studies to demonstrate that ITAM and CMAM can be trained with smaller models and perform well without pre-trained models.
   - Validated the benefits of cross-modality fusion in CMAM and the necessity of SMM for temporal coherence.

3. **Efficiency Analysis**:
   - Added runtime analyses in Appendix H, demonstrating competitive performance with an average frame rate of 12 FPS using YOLO-v7 for object detection.

4. **Generalization Improvements**:
   - Highlighted how local patterns generalize better across domains by capturing semantically meaningful components like body joints.
   - Addressed low-resolution scenarios by refining captions with SMM, ensuring consistent performance.

5. **Framework Simplification**:
   - Demonstrated that smaller models (e.g., YOLO-v7) achieve similar performance to larger models (e.g., Qwen-VL-7B), reducing dependency on heavy pre-trained models.

---

## **Remaining Weaknesses**
1. **Framework Complexity**:
   - Despite clarifications, the overall framework remains complex, which may deter practitioners unfamiliar with advanced multimodal techniques.

2. **Reliance on Vision-Language Models**:
   - Although the authors addressed concerns through ablation studies, the use of pre-trained models for generating training labels introduces reliance that could be reduced further.

3. **Broader Applicability**:
   - The framework focuses on benchmark datasets but lacks validation in diverse real-world scenarios, such as highly cluttered environments or surveillance with significant occlusions.

---

## **Justification for Acceptance**
This paper makes a significant contribution to video anomaly detection by introducing a novel approach that prioritizes local patterns, which are more generalizable to novel anomalies. The combination of ITAM, CMAM, and SMM provides a robust framework that effectively integrates spatial and temporal reasoning. The authors' revisions address most reviewer concerns, improving clarity, reducing reliance on pre-trained models, and demonstrating the efficiency of their approach.

While some complexity remains, the paper's methodological innovation, strong experimental results, and community contributions (e.g., open-source code) outweigh these limitations. This work has the potential to advance the field of anomaly detection and inspire further research. I recommend acceptance.

**Additional Comments On Reviewer Discussion:**

Please refer to details in the above section

---

### Decision · Program_Chairs · 2025-01-22

Accept (Poster)